

# A new approach for SPN removal: nearest value based mean filter

Bülent Turan

Department of Computer Engineering/Faculty of Engineering and Architecture, Tokat Gaziosmanpasa University, Tokat, Turkey

## ABSTRACT

In this study, a new adaptive filter is proposed to eliminate salt and pepper noise (SPN). The basis of the proposed method consists of two-stages. (1) Changing the noisy pixel value with the closest pixel value or assigning their average to the noisy pixel in case there is more than one pixel with the same distance; (2) the updating of the calculated noisy pixel values with the average filter by correlating them with the noise ratio. The method developed was named as Nearest Value Based Mean Filter (NVBMF), because of using the pixel value which the closest distance in the first stage. Results obtained with the proposed method: it has been compared with the results obtained with the Adaptive Frequency Median Filter, Adaptive Riesz Mean Filter, Improved Adaptive Weighted Mean Filter, Adaptive Switching Weight Mean Filter, Adaptive Weighted Mean Filter, Different Applied Median Filter, Iterative Mean Filter, Two-Stage Filter, Multistage Selective Convolution Filter, Different Adaptive Modified Riesz Mean Filter, Stationary Framelet Transform Based Filter and A New Type Adaptive Median Filter methods. In the comparison phase, nine different noise levels were applied to the original images. Denoised images were compared using Peak Signal-to-Noise Ratio, Image Enhancement Factor, and Structural Similarity Index Map image quality metrics. Comparisons were made using three separate image datasets and Cameraman, Airplane images. NVBMF achieved the best result in 52 out of 84 comparisons for PSNR, best in 47 out of 84 comparisons for SSIM, and best in 36 out of 84 comparisons for IEF. In addition, values nearly to the best result were obtained in comparisons where the best result could not be reached. The results obtained show that the NVBMF can be used as an effective method in denoising SPN.

## INTRODUCTION

Image processing is one of the most studied areas in recent years. Development of computer technologies: It allows to increase the processing speed and therefore to perform complex operations with large data sizes. Images have large data sizes and carry a lot of information. For this reason, nowadays, subjects such as feature extraction, object detection, object tracking, image classification, pattern recognition are intensively studied in addition to classical image processing (image processing for visual purposes, image compression, *etc.*).

Corresponding author
Bülent Turan,
bulent.turan@gop.edu.tr

Image denoise is an important pre-processing in image processing applications (*Loya & Keskar, 2015*; *Erkan et al., 2020a*; *Chen, Hung & Zou, 2017*; *Wang et al., 2016a*; *Erkan et al., 2019*). Image processing techniques such as segmentation, feature extraction, morphological processes etc. cannot be effectively applied on the noisy image. As a result, object detection, tracking, image classification, pattern recognition, *etc.*, which are popular study areas of today, cannot be carried out in a stable manner on images that contain raw noise and are difficult to analyze. For this reason, noise removal, is still a popular field of study.

Noises usually occur during the create and transmission of the image. One of these noises is SPN. SPN denoising is a popular field of study. There are many studies in SPN denoising (*Loya & Keskar, 2015*; *Erkan et al., 2020a*; *Chen, Hung & Zou, 2017*; *Wang et al., 2016b*; *Erkan et al., 2019*; *Wang et al., 2016a*; *Zhang & Li, 2014*; *Erkan et al., 2020b*; *Thanh et al., 2020a*; *Enginoğlu, Erkan & Memiş, 2020*; *Memis & Erkan, 2021*; *Enginoğlu, Erkan & Memiş, 2019*; *Thanh et al., 2020b*; *Yildiz & Yildiz, 2018*; *Gökcen & Kalyoncu, 2020*; *Erkan & Gökrem, 2018*; *Duan & Zhang, 2010*; *Erkan, Gökrem & Enginoğlu, 2018*; *Erkan & Gökrem, 2017*). The SPN is cleaned with spatial filters in the image. Average, weighted average and other averaging filters have better results when the noisy pixel values are calculated *via* the average of the pixel values without noise (*Wang et al., 2016b*; *Erkan et al., 2019*; *Wang et al., 2016a*; *Zhang & Li, 2014*; *Erkan et al., 2020b*; *Thanh et al., 2020a*; *Enginoğlu, Erkan & Memiş, 2020*; *Memis & Erkan, 2021*; *Enginoğlu, Erkan & Memiş, 2019*; *Xu & Aminu, 2022*; *Hien et al., 2022*, *Bindal & Garg, 2022*; *Aslam et al., 2022*). Due to the linearity of these filters, blurring occurs while improving the image. On the other hand, median filters are nonlinear filters and do not cause blurring of edges. They are also widely used to remove the SPN (*Erkan et al., 2020a*; *Duan & Zhang, 2010*; *Erkan, Gökrem & Enginoğlu, 2018*; *Erkan & Gökrem, 2017*; *Erkan, Gökrem & Enginoğlu, 2019*; *Sharadqh et al., 2019*; *Yildirim, 2021*). However, even if the median filters take the median of only the noiseless pixels, they do not provide good results because they cannot express the neighbourhood relationship at a high-level noise (*Loya & Keskar, 2015*).

In this study, the "Nearest Value Based Mean Filter (NVBMF)" method is proposed to remove SPN from the image. NVBMF is compared with state-of-the-art SPN filter methods. These are the Adaptive Frequency Median Filter (AFMF), (*Erkan et al., 2020a*), Adaptive Riesz Mean Filter (ARmF) (*Enginoğlu, Erkan & Memiş, 2019*), Improved Adaptive Weighted Mean Filter (IAWMF) (*Erkan et al., 2020b*), Adaptive Switching Weight Mean Filter (ASWMF) (*Thanh et al., 2020a*), Adaptive Weighted Mean Filter (AWMF) (*Zhang & Li, 2014*), Different Applied Median Filter (DAMF) (*Erkan, Gökrem & Enginoğlu, 2018*), Iterative Mean Filter (IMF) (*Erkan et al., 2019*), Two-Stage Filter (TSF) (*Thanh et al., 2020b*), Multistage Selective Convolution Filter (MSCF-1) (*Rafiee & Farhang, 2022*), Different Adaptive Modified Riesz Mean Filter (DAMRmF) (*Memis & Erkan, 2021*), Stationary Framelet Transform Based Filter (SFT_lp) (*Chen et al., 2022*) and A New Type Adaptive Median Filter (Bilal's Method) (*Charmouti et al., 2022*). Results are compared by using image quality metrics: Peak Signal-to-Noise Ratio (PSNR) (*Enginoğlu, Erkan & Memiş, 2019*; *Turan, 2021*; *Olmez, Sengur & Özmen Koca, 2020*), Image Enhancement Factor (IEF) (*Enginoğlu, Erkan & Memiş, 2019*; *Djurovi¢, 2017*) and

Structural Similarity (SSIM) (*Turan, 2021*; *Olmez, Sengur & Özmen Koca, 2020*; *Wang et al., 2004*).

# PROPOSED DENOISING METHOD

## Definitions and notions

This paper, let $f(x, y) = [f_{ij}]_{m \times n}$ be an original image, $f'(x, y) = [f'_{ij}]_{m \times n}$ restored image, $y(x, y) = [y_{ij}]_{m \times n}$ 3$^{th}$ stage result image, $g(x, y) = [g_{ij}]_{m \times n}$ salt and pepper noisy image, and $g'(x, y) \equiv g(x, y) (mod\ 255)$ hence only pepper noisy image in $g'(x, y) = [g'_{ij}]_{m \times n}$ in $m \times n$ sizes.

**Definition 1**. In the gray level images, the smallest value ($\gamma_{min}$) is 0, and the largest value ($\gamma_{max}$) is 255. SPN (salt and pepper noisy) and PN (pepper noisy) models with the same noise levels as follows:

$$g_{ij} = \begin{cases} \gamma_{min}, & \textbf{with probability } \textbf{p} \\ \gamma_{max}, & \textbf{with probability q} \\ f_{ij}, & \textbf{with probability } \textbf{1} - (\textbf{p} + \textbf{q}) \end{cases} \cdots$$

$$g'_{ij} = g_{ij}(mod\ 255) \cdots$$

$$g'_{ij} = \begin{cases} \gamma_{min}, & \textit{with probability } p + q \\ f_{ij}, & \textit{with probability } \textbf{1} - (p + q) \end{cases} \tag{1}$$

**Definition 2**. $N$L is noisy levels. Noise level is the ratio of the number of zeros to the total number of pixels in $g'_{ij}$. $Z_{ij}$ binary matrix in $m \times n$ sizes.

$$Z_{ij} = \begin{cases} \textbf{1}, & g'_{ij} = \textbf{0} \\ \textbf{0}, & g'_{ij} \neq \textbf{0} \end{cases} \tag{2}$$

$$NL = \frac{\sum_{ij} Z_{ij}}{m \times n} \tag{3}$$

**Definition 3**. $S_{ij}(w)$ be a image window in $[g'_{ij}]_{m \times n}$. It is size of $11 \times 11$ with central pixel coordinates (i,j). The pixel value with the smallest Euclidean distance or average of pixels with the same smalest Euclidean distance (**npv**).

$$S^{npv}_{ij}(w) = \begin{cases} K_{ij}(w), & g'_{ij} = \textbf{0} \\ g'_{ij}, & \textit{otherwise} \end{cases} \tag{4}$$

$$\delta_{i^*j^*} = \begin{cases} \sqrt{(i - i^*)^2 + (j - j^*)^2}, & S_{ij} \neq \textbf{0} \\ \varnothing, & S_{ij} = \textbf{0} \end{cases} \tag{5}$$

$$K_{ij}(w) = \begin{cases} g'_{i^*j^*}, & \textit{If there is only one pixel with distance } \delta^{min}_{ij}(w) \\ g'^{Mean}_{i^*j^*}, & \textit{If there is too many pixels with distance } \delta^{min}_{ij}(w) \\ y^{Mean}_{ij}, & \textit{If there is no pixels with distance } \delta^{min}_{ij}(w) \end{cases} \tag{6}$$

**Definition 4**. $\beta^{Mean}_{ij}(w)$ be a image window in $[y_{ij}]_{m \times n}$. It is size of $3 \times 3$ with central pixel coordinates (i,j). $R_{ij}$ is the average of nonzeros $y_{i^*j^*}$ values.

$$\beta_{ij}^{Mean}(w) = \begin{cases} R_{ij}(w), & NL > 0.45 \\ y_{ij}, & otherwise \end{cases} \tag{7}$$

$$R_{ij}(w) = \frac{\sum_{(i^*j^*) \in \beta_{ij}(w)} y_{i^*j^*}}{\sum_{(i^*j^*) \in \beta_{ij}(w)} D_{i^*j^*}} \tag{8}$$

$$D_{i^*j^*} = \begin{cases} 1, & y_{i^*j^*} \neq 0 \\ 0, & y_{i^*j^*} = 0 \end{cases} \tag{9}$$

## Proposed method

The IAWMF method is the development of the AWMF method of determining the weights. In this method, the weights of the noisy pixels are taken to 0, while the weights of the noisy pixels are associated with the Euclidean distance to the pixel to be filtered. Thus, the weight of the neighbouring pixel value nearly to the pixel to be filtered is provided to be higher. In this method, the weight values are determined by Eq. (10).

$$D_{i^*j^*} = \frac{1}{\left( \varepsilon + \sqrt{(i - i^*)^2 + (j - j^*)^2} \right)^4}, \ 0 < \varepsilon \ll 1 \tag{10}$$

It is a positive approach to determine the weights by Euclidean distance in the IAWMF method. However, the association of weight with the fourth power of Euclidean distance causes the weights of long-distance pixels to get too small. Therefore, pixel values at the nearest distance predominantly affect the pixel value to be filtered.

The method suggested in the study is based on the calculation of the pixel value to be filtered in two stages. In the first stage, the nearest noiseless pixel value is assigned to the pixel to be filtered. If there is more than one pixel with the same distance value, their averages are assigned as pixels values. At this stage, the filter size $11 \times 11$ is used. The reason for this is to guarantee the presence of noiseless pixels in the filter even at high level noise. However, because noise is randomly distributed throughout the image, even a large filter size may sometimes not guarantee noiseless pixels in the frame. In this case, it is assigned as the pixel value to be filtered by taking the average of the previously filtered pixel values.

Images with a low level noise usually have multiple pixel with the lowest Euclidean distance. Thus, the noisy pixel value in the first stage is calculated by taking the average of these pixels. As the noise ratio in the image increases, the number of pixels with the lowest Euclidean distance will decrease; whereas at high levels of noise, there will usually be only one pixel with the lowest Euclidean distance. In this case, this value will be assigned to the noisy pixel.

In the first stage, the closest pixel value is usually assigned without averaging in images with a high noise ratio. Although this value is close to the original value, this prevents the image from reaching sufficient quality in terms of resolution. For this reason, in cases where the noise ratio is above 45%, the second stage is applied after the first stage. At second stage, an average filter at $3 \times 3$ size is applied. Pixels with the value of 0 are ignored

**Algorithm 1** Nearest Value Based Mean Filter (NVBMF).

**Input:** A noisy image $\quad g(x, y) := [g_{ij}]_{m \times n}$

**Output:** A restored image $\quad f'(x, y) := [f'_{ij}]_{m \times n}$

Initialize $\ g'(x, y) = g(x, y)(mod\ 255)$

Compute $\ Z_{ij}$, NL

**For** each pixel of PN noisy image $[g'_{ij}]_{m \times n}$

  **If** $g'_{ij} \neq 0$

    $y_{ij} = g'_{ij}$

  **Else**

    $y_{ij} = S^{npv}_{ij}(w)$

  **End**

**End**

**For** each pixel of the image $[y_{ij}]_{mxn}$

  **If** NL>0.45

    **If** $g'_{ij} \neq 0$

      $f'_{ij} = g'_{ij}$

    **Else**

      $f'_{ij} = \beta^{Mean}_{ij}(w)$

    **End**

  **Else**

    $f'_{ij} = g'_{ij}$

  **End**

**End**

when applying the Average filter. The algorithm and flow chart of the proposed method is given below (Fig. 1).

# EXPERIMENTAL RESULTS AND DISCUSSIONS

## Quality metrics

In this section, image quality metrics are provided in order to compare denoising filters used for salt and peppers noisy image. Peak Signal-to-Noise Ratio (PSNR), Structural Similarity (SSIM) and Image Enhancement Factor (IEF) were used in the study to evaluate the image quality. PSNR and MSE (*Enginoğlu, Erkan & Memiş, 2019*; *Turan, 2021*; *Olmez, Sengur & Özmen Koca, 2020*) defined as:

$$PSNR(X, Y) = 10log_{10}\left[\frac{(L-1)^2}{MSE}\right] \tag{11}$$

$$MSE(X, Y) = \frac{1}{MN}\sum_{i=1}^{i=M}\sum_{j=1}^{j=N}[X(i,j) - Y(i,j)]^2 \tag{12}$$

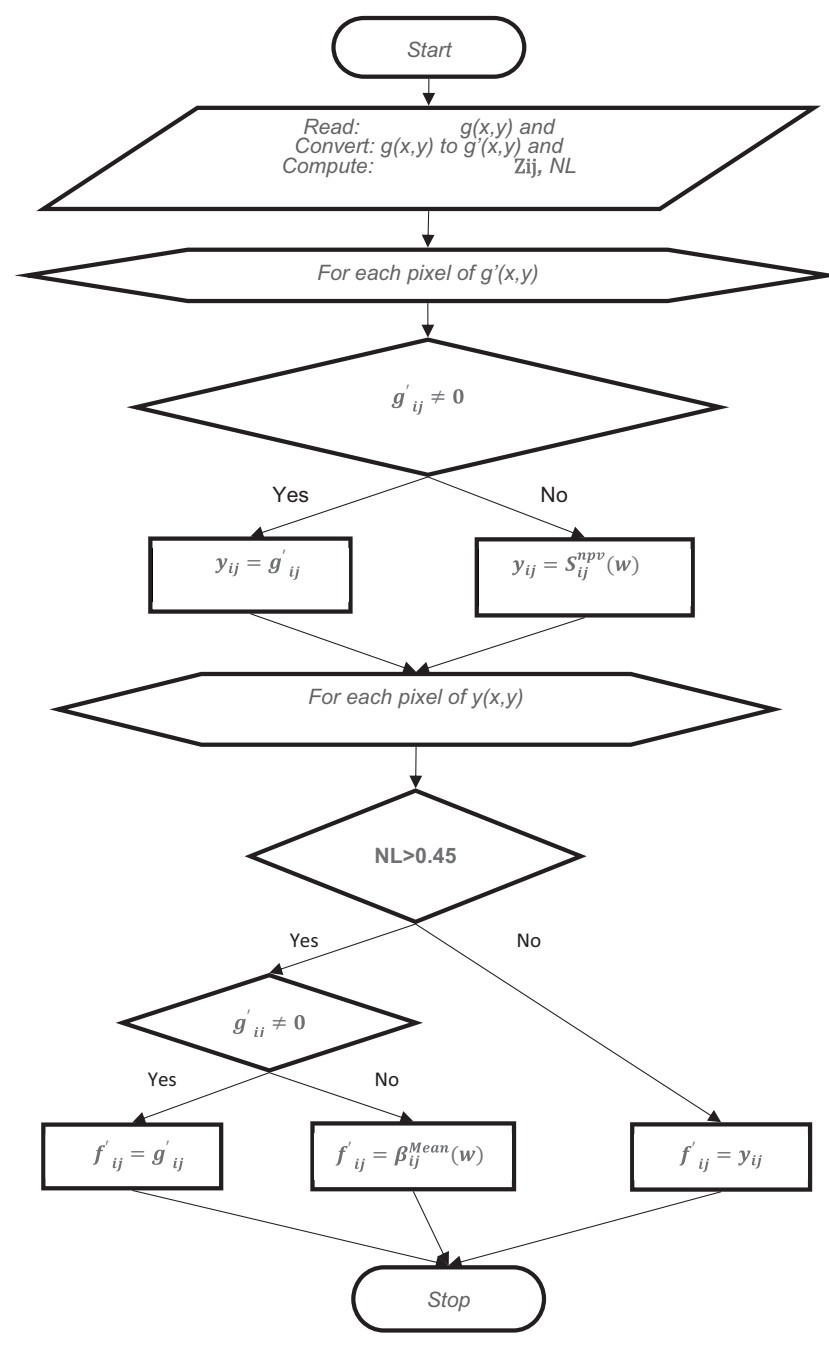

**Figure 1 The flowchart of NVBMF.**

SSIM (*Turan, 2021*; *Olmez, Sengur & Özmen Koca, 2020*; *Wang et al., 2004*) defined as:

$$SSIM(X, Y) = [I(X, Y)]^{\alpha} * [c(X, Y)]^{\beta} * [s(X, Y)]^{\gamma} \tag{13}$$

$$I(X, Y) = \frac{2\mu_X \mu_{Y+} C_1}{\mu_X^2 + \mu_Y^2 + C_1} \tag{14}$$

$$c(X, Y) = \frac{2\sigma_X \sigma_{Y+} C_2}{\sigma_X^2 + \sigma_Y^2 + C_2} \tag{15}$$

$$s(X, Y) = \frac{\sigma_{XY} + C_3}{\sigma_X \sigma_Y + C_3} \tag{16}$$

If $\alpha = \beta = \gamma = 1$ and $C_3 = C_2/2$, $\Rightarrow SSIM(X, Y)$ is organized as follows.

$$SSIM(X, Y) = \frac{(2\mu_X \mu_{Y+} C_1)(2\sigma_{XY} + C_2)}{(\mu_X^2 + \mu_Y^2 + C_1)(\sigma_X^2 + \sigma_Y^2 + C_2)} \tag{17}$$

IEF (*Enginoğlu, Erkan & Memiş, 2019*; *Djurović, 2017*) defined as:

$$IEF(X, Y, Z) = \frac{\sum_{i=1}^{M} \sum_{j=1}^{N} (z_{ij} - x_{ij})^2}{\sum_{i=1}^{M} \sum_{j=1}^{N} (y_{ij} - x_{ij})^2} \tag{18}$$

In the above equations, $X = [x_{ij}]$ is an original image, $Y = [y_{ij}]$ is denoised image, $Z = [z_{ij}]$ is a noisy image.

## Image databases

In the study, three different image datasets are used to compare the methods:

1. *UC-Berkeley dataset (BSDS)—200 images* (*Arbelaez, Fowlkes & Martin, 2007*);
2. *TESTIMAGES dataset—40 images* (*Asuni & Giachetti, 2015*);
3. *MATLAB library images—20 images (R2020b; autumn, baby, board, micromarket, car1, coloredChips, fabric, foggyroad, foggysf1, foosball, football, greens, gantrycrane, trailer, hallway, hands1, pears, kobi, lighthouse, onion).*

MATLAB Library images can be accessed from a computer with the MATLAB R2020b program installed using the given directory (MATLAB\R2020b\toolbox\images\imdata).

The proposed method and other methods were compared using datasets. Noise was added to each image at the ratios of 0.1, 0.2, 0.3, 0.4, 0.5, 0.6, 0.7, 0.8, 0.9. For example, 1,800 PSNR, SSIM, and IEF results were obtained for BSDS. The average of all levels of noise was calculated separately for each image quality metric (PSNR, SSIM, IEF). In addition, the general average of all levels of noise has been added to the tables.

In addition, all methods were compared for six images (three from datasets and three from outside of datasets). Thus, it was ensured that the individual performances of all methods on images were observed.

## DISCUSSIONS

In the first test phase, images of Cameraman and jelly beans (4.1.07) (https://dome.mit.edu/handle/1721.3/195767, https://sipi.usc.edu/database/database.php?volume=misc#top) were used. In this test stage, jelly beans (4.1.07) was used with 90% and 60% levels of noise, Cameraman 60% levels of noise, and the noisy images were denoised with NVBMF and other methods. NVBMF for denoising the 90% noisy jelly beans (4.1.07) image were

obtained the best result in both PSNR and SSIM comparison than other methods (Fig. 2). NVBMF for denoising the 60% noisy Cameraman image were obtained the best result in the PSNR comparison and the third-best result in the SSIM comparison (Fig. 3). NVBMF for denoising 60% noisy jelly beans (4.1.07) image were obtained the best result in both PSNR and SSIM comparison (Fig. 4). In addition, eight different noisy Cameraman images from 10% to 80% were denoising with NVBMF, and the results are given in Fig. 5. Also, the proposed method has been tested with color images. eight different noisy jelly beans (4.1.07) images from 20% to 90% were denoising with NVBMF, and the results are given in Fig. 6.

In the second test phase, six images (Lena, Cameraman, Airplane, Micromarket, img_600 × 600_1 × 8 bit_B01C00GRAY_apples, 177083) were compared in all noise ratios. In this comparison, one image from each dataset and three images excluding the datasets were selected. Results are provided in Tables 1–3. Out of the 27 comparisons made for the Lena image, 17 of them reached the best result. The best results were obtained in seven out of nine comparisons for PSNR, six out of nine comparisons for SSIM, and four out of nine comparisons for IEF. The best result was achieved in 16 out of 27 comparisons made for the cameraman image. The best results were obtained in six out of nine comparisons for PSNR, five out of nine comparisons for SSIM, and five out of nine comparisons for IEF. The best results were achieved in 19 out of 27 comparisons made for the airplane image. The best results were obtained in six out of nine comparisons for PSNR, seven out of nine comparisons for SSIM, and six out of nine comparisons for IEF. The best results were achieved in 11 of the 27 comparisons made for the Micromarket image. The best results were obtained in four out of nine comparisons for PSNR, four out of nine comparisons for SSIM, and three out of nine comparisons for IEF. The best results were achieved in 16 out of 27 comparisons made for img_600 × 600_1 × 8 bit_B01C00GRAY_apples image. The best results were obtained in four out of nine comparisons for PSNR, seven out of nine comparisons for SSIM, and five out of nine comparisons for IEF. The best results were achieved in 11 out of 27 comparisons made for 187,083 images. The best results were obtained in three out of nine comparisons for PSNR, six out of nine comparisons for SSIM, and two out of nine comparisons for IEF.

The last test phase was completed on datasets. At this stage, noise from 10% to 90% has been added to the dataset images. Thus, nine noisy images with different levels of noise were obtained for each dataset image. Each of the noisy images was denoised with the compared noise removal methods and the results were averaged.

The results obtained from 1,800 cleaning processes for BSDS are given in Tables 4–6. NVBMF obtained the best results in the comparisons for PSNR with 10%, 20%, 30%, 60%, 70%, 80% levels of noise averages and general average values. For 40%, 50%, and 90% levels of noise, 3rd, 3rd, and 3rd. were obtained the best results, respectively (Table 4). NVBMF also obtained the best results in the comparisons for SSIM with the average values of 10%, 20%, 30%, 80% levels of noise. For 40%, 50%, 60%, 70%, 90% and general levels of noise, the 3rd, 5rd, 4rd, 3nd, 4rd, 5rd respectively were obtained the best results (Table 5). NVBMF also obtained the best results in the comparisons for IEF with 10%, 70%, 80%,

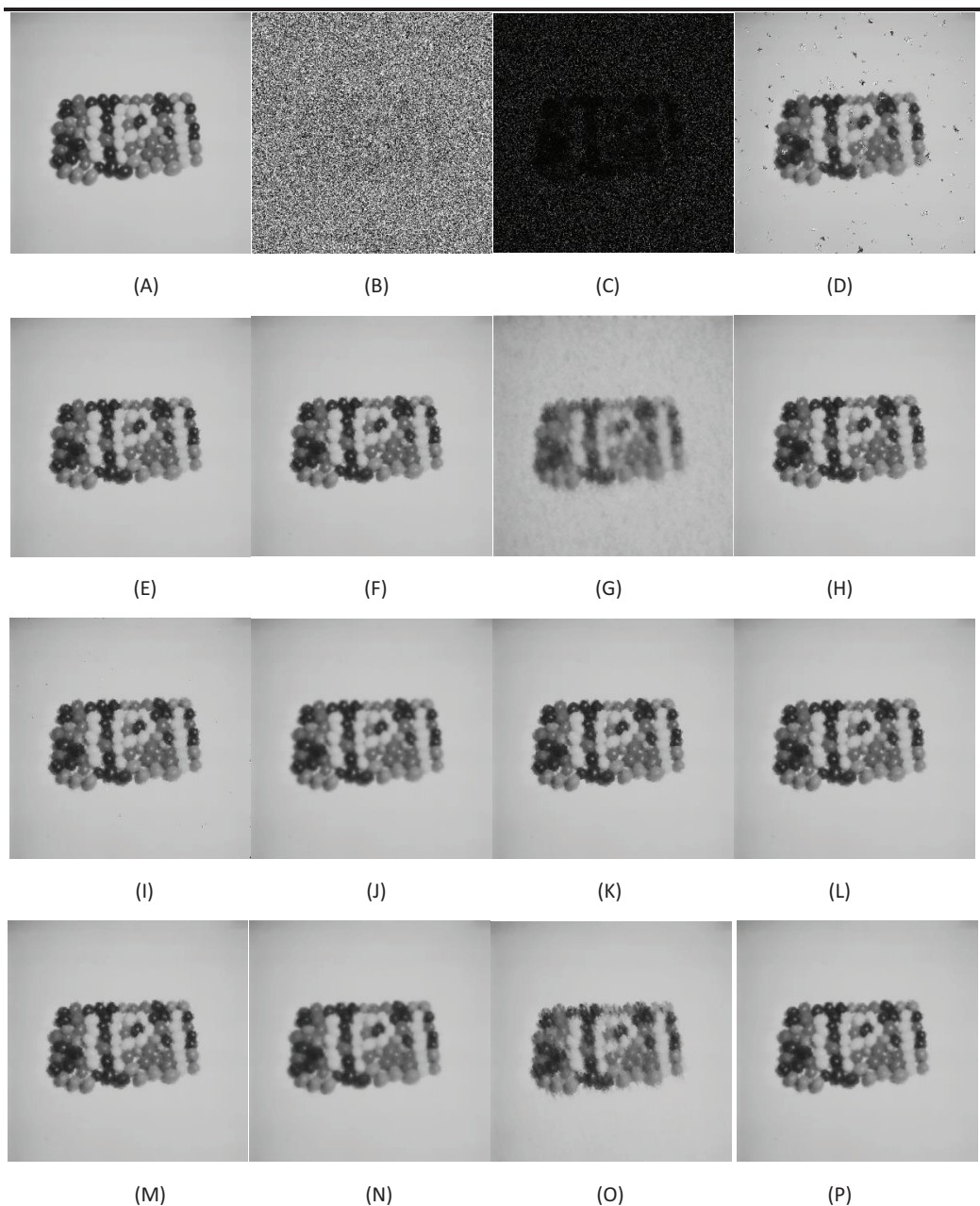

**Figure 2 Denoising results by different filters for the jelly beans (4.1.07) with the size of 512 × 512 pixels with 90% SPN levels.** (A) Original image, (B) noise of 90% (SPN) PSNR = 5.55 SSIM = 0.0044, (C) noise of 90% (PN) PSNR = 3.48 SSIM = 0.0062, (D) denoising AFMF PSNR = 23.54 SSIM = 0.8623, (E) denoising ARmF PSNR = 33.24 SSIM = 0.9577, (F) denoising IAWMF PSNR = 34.03 SSIM = 0.9639, (G) denoising ASWMF PSNR = 23.88 SSIM = 0.7462, (H) denoising AWMF PSNR = 33.19 SSIM = 0.9571, (I) denoising DAMF PSNR = 31.89 SSIM = 0.9487, (J) denoising IMF PSNR = 35.06 SSIM = 0.9727, (K) denoising TSF PSNR = 32.87 SSIM = 0.9559, (L) denoising MSCF-1 PSNR = 34.30 SSIM = 0.9654, (M) denoising DAMRmF PSNR = 34.19 SSIM = 0.9654, (N) denoising SFT_lp PSNR = 33.73 SSIM = 0.9665, (O) denoising Bilal's method PSNR = 26.76 SSIM = 0.9063, (P) denoising NVBMF (proposed) PSNR = 35.60 SSIM = 0.9744.

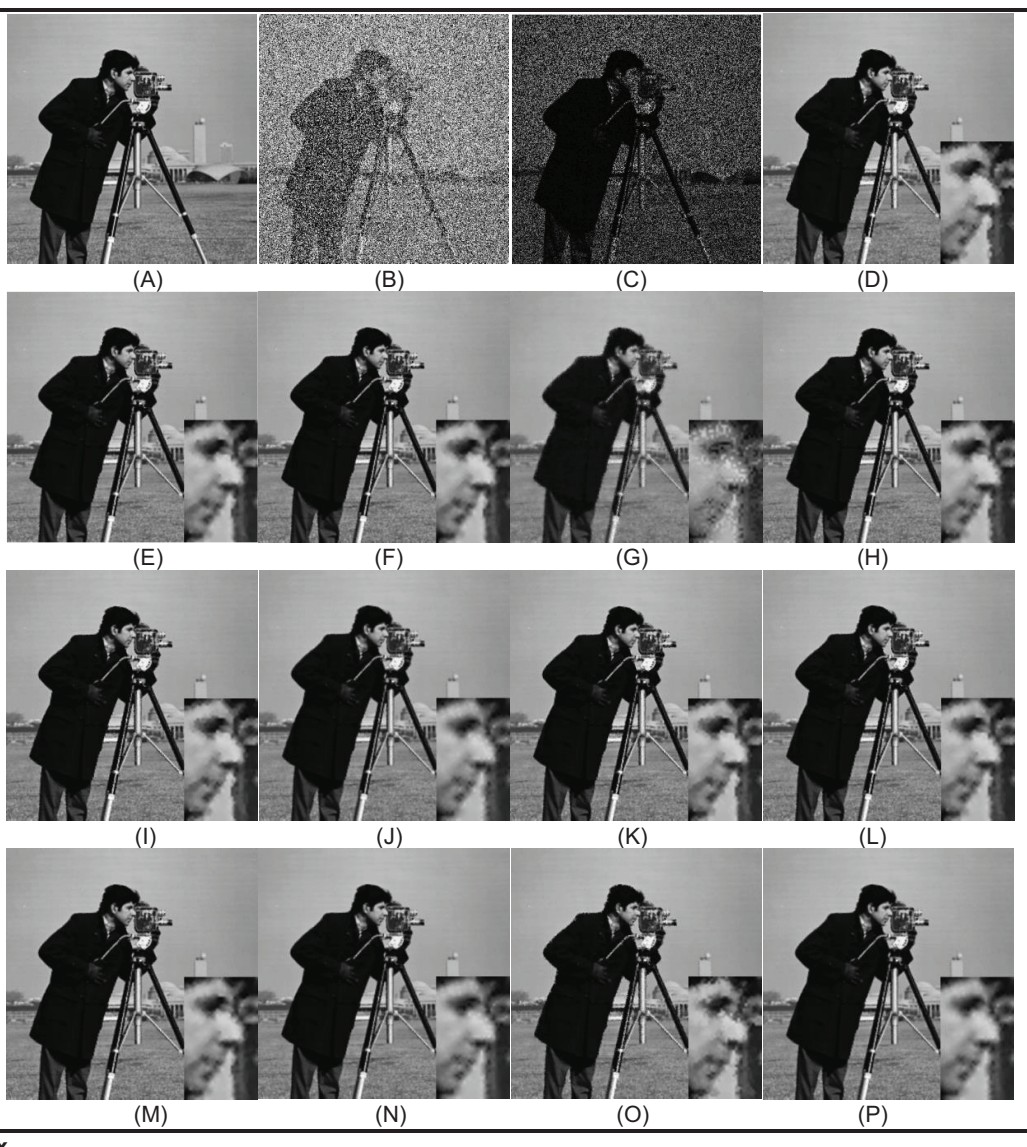

**Figure 3 Denoising results by different filters for the Cameraman with the size of 512 × 512 pixels with 60% SPN levels.** (A) Original image, (B) noise of 60% (SPN) PSNR = 7.31 SSIM = 0.0204, (C) noise of 60% (PN) PSNR = 7.82 SSIM = 0.1107, (D) denoising AFMF PSNR = 30.10 SSIM = 0.9393, (E) denoising ARmF PSNR = 32.62 SSIM = 0.9630, (F) denoising IAWMF PSNR = 32.88 SSIM = 0.9646, (G) denoising ASWMF PSNR = 26.63 SSIM = 0.8406, (H) denoising AWMF PSNR = 32.12 SSIM = 0.9580, (I) denoising DAMF PSNR = 31.40 SSIM = 0.9541, (J) denoising IMF PSNR = 32.46 SSIM = 0.9558, (K) denoising TSF PSNR = 31.38 SSIM = 0.9540, (L) denoising MSCF-1 PSNRS = 32.35 SSIMS = 0.9606, (M) denoising DAMRmF PSNR = 32.86 SSIM = 0.9634, (N) denoising SFT_lp PSNR = 32.49 SSIM = 0.9563, (O) denoising Bilal's method PSNR = 28.74 SSIM = 0.9193, (P) denoising NVBMF (proposed) PSNR = 33.36 SSIM = 0,9630. Image source: CC BY NC; https://hdl.handle.net/1721.3/195767.

90% and general average values. They were obtained the 2rd, 5rd, 5rd, 2rd, 2rd best results for 20%, 30% 40%, 50% and 60% levels of noise, respectively (Table 6).

Results obtained from 360 denoising operations for TESTIMAGES dataset are given in Tables 7–9. NVBMF obtained the best results in the comparisons for PSNR with 60%, 70%,

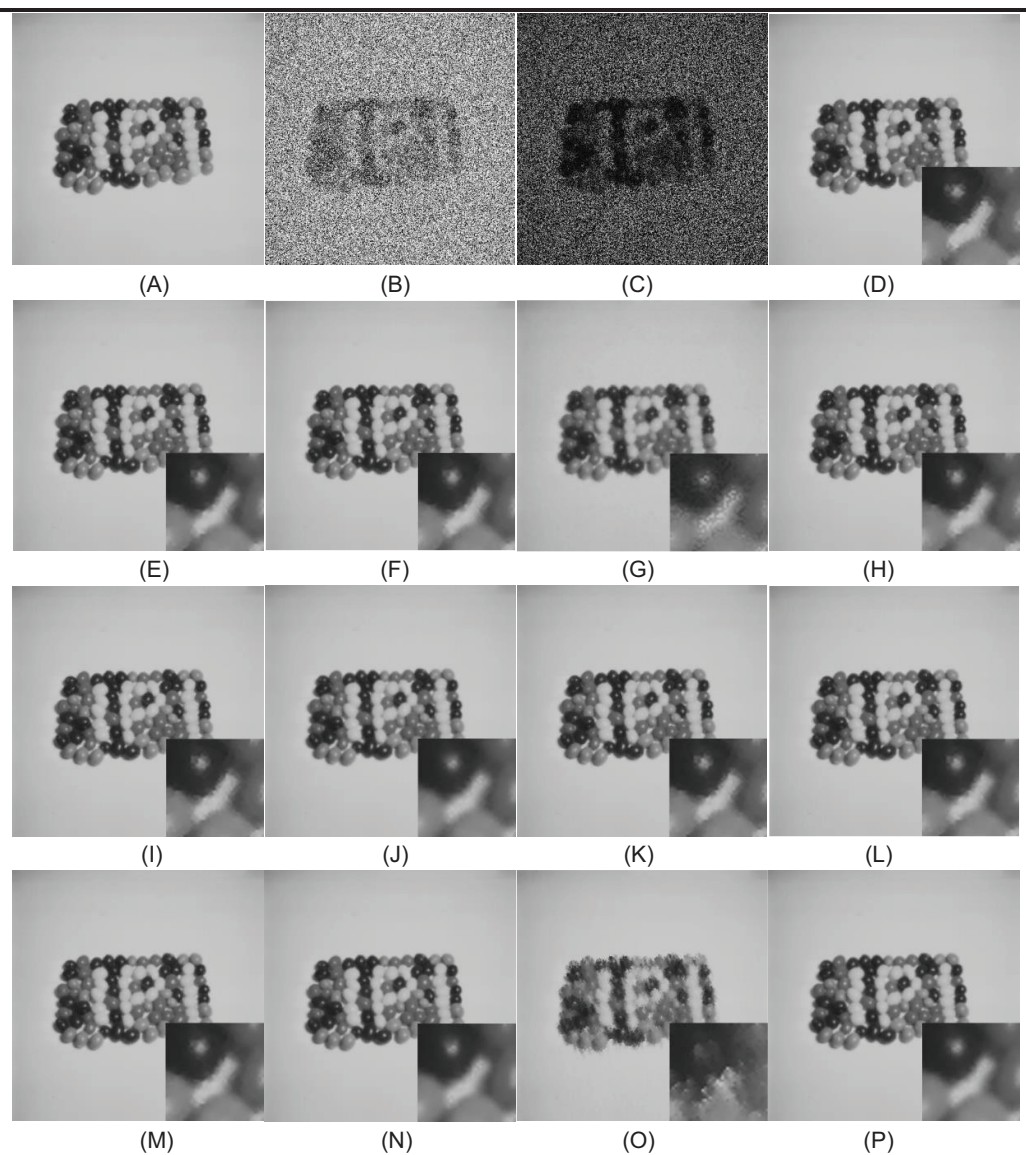

**Figure 4 Denoising results by different filters for the Jelly beans (4.1.07) with the size of 512 × 512 pixels with 60% SPN levels.** (A) Original image, (B) noise of 60% (SPN) PSNR = 7.34 SSIM = 0.0097, (C) noise of 60% (PN) PSNR = 5.25 SSIM = 0.0151, (D) denoising AFMF PSNR = 39.04 SSIM = 0.9856, (E) denoising ARmF PSNR = 41.84 SSIM = 0.9916, (F) denoising IAWMF PSNR = 42.17 SSIM = 0.9921, (G) denoising ASWMF PSNR = 33.81 SSIM = 0.9332, (H) denoising AWMF PSNR = 41.42 SSIM = 0.9907, (I) denoising DAMF PSNR = 40.32 SSIM = 0.9888, (J) denoising IMF PSNR = 43.45 SSIM = 0.9936, (K) denoising TSF PSNR = 29.13 SSIM = 0.9244, (L) denoising MSCF-1 PSNR = 41.61 SSIM = 0.9912, (M) denoising DAMRmF PSNR = 42.44 SSIM = 0.9925, (N) denoising SFT_lp PSNR = 44.28 SSIM = 0.9944, (O) Bilal's method PSNR = 26.76 SSIM = 0.9063, (P) denoising NVBMF (proposed) PSNR = 44.50 SSIM = 0.9947.

80% levels noise averages values. For 10%, 20% 30%, 40%, 50% 90% levels of noise and general average values, 2rd, 2rd, 4rd, 6rd, 2rd, 2rd, and 2rd were obtained the best results, respectively (Table 7). NVBMF also obtained the best results in the comparisons for SSIM with the average values of 10%, 20% levels noise. For 30%, 40%, 50%, 60%, 70%, 80% 90%

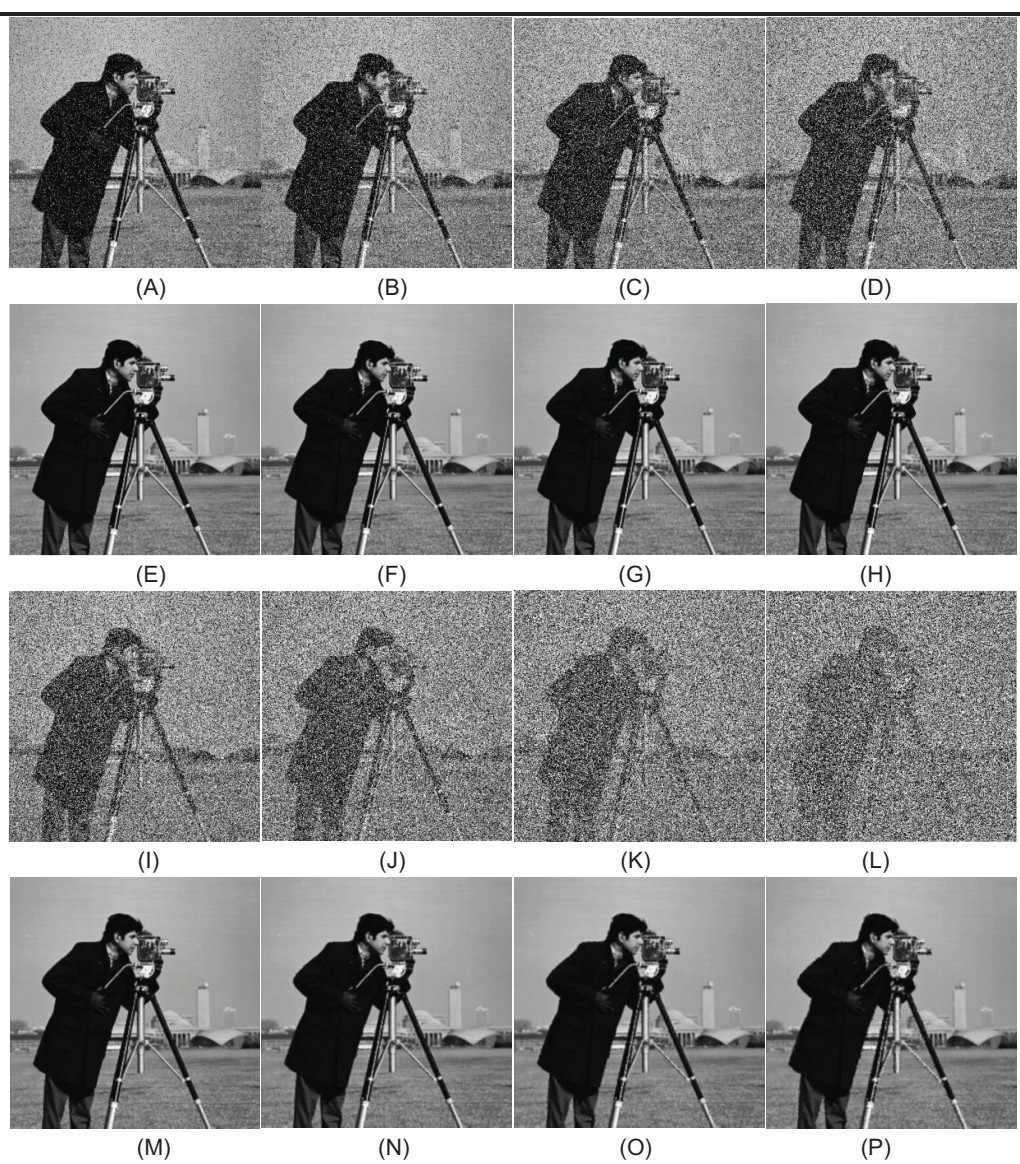

**Figure 5** **Denoising results by proposed method (NVBMF) on Cameraman image.** (A) Noisy image (10%), (B) noisy image (20%), (C) noisy image (30%), (D) noisy image (40%), (E) repaired image (10%) PSNR = 45.50 SSIM = 0.9948, (F) repaired image (20%) PSNR = 41.33 SSIM = 0.9943, (G) repaired image (30%) PSNR = 38.35 SSIM = 0.9895, (H) repaired image (40%) PSNR = 35.90 SSIM = 0.9820, (I) noisy image (50%), (J) noisy image (60%), (K) noisy image (70%), (L) noisy image (80%), (M) repaired image (50%) PSNR = 34.94 SSIM = 0.9736, (N) repaired image (60%) PSNR = 33.36 SSIM = 0.9630, (O) repaired image (70%) PSNR = 31.44 SSIM = 0.9480, (P) repaired image (80%) PSNR = 29.57 SSIM = 0.9226. Image source: CC BY NC; https://hdl.handle.net/1721.3/195767.

and general levels of noise, 2rd, 4rd, 4rd, 8rd, 8rd, 4rd, 6rd, 6rd were obtained the best results, respectively (Table 8). NVBMF also obtained the best results in the comparisons for IEF with 70%, 80%, 90% averages of levels of noise values. For 10%, 20%, 30%, 40%, 50%, 60% levels noise and general average values, 3rd, 4rd, 6rd, 6rd, 2rd, 2rd and 2rd were obtained the best results, respectively (Table 9).

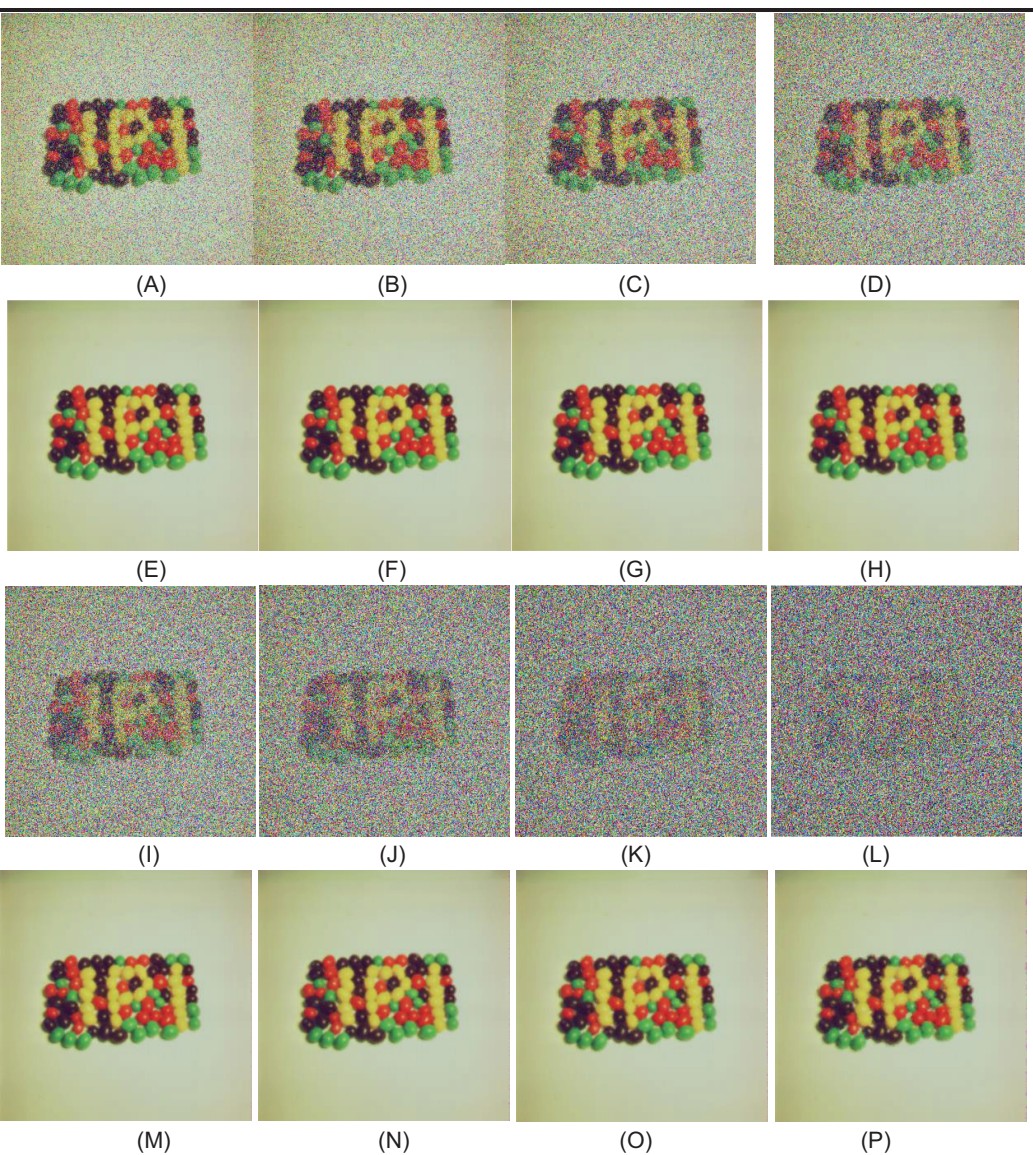

**Figure 6 Denoising results by proposed method (NVBMF) on color jelly beans (4.1.07) image.**
(A) Noisy image (20%), (B) noisy image (30%), (C) noisy image (40%), (D) noisy image (50%),
(E) repaired image (20%) PSNR = 50.24 SSIM = 0.9995, (F) repaired image (30%) PSNR = 46.50 SSIM =
0.9990, (G) repaired image (40%) PSNR = 44,13 SSIM = 0.9984, (H) repaired image (50%) PSNR = 44.58
SSIM = 0.9982, (I) noisy image (60%), (J) noisy image (70%), (K) noisy image (80%), (L) noisy image
(90%), (M) repaired image (60%) PSNR = 42.99 SSIM = 0.9977, (N) repaired image (70%) PSNR = 41.13
SSIM = 0.9966, (O) repaired image (80%) PSNR = 38.67 SSIM = 0.9947, (P) repaired image (90%) PSNR
= 34.90 SSIM = 0.9889.

The results obtained from 180 cleaning processes for the dataset created for MATLAB
images are given in Tables 10–12. In all comparisons for PSNR, NVBMF were obtained the
best result (Table 10). NVBMF obtained the best results in the comparisons for SSIM
with 10%, 20%, 30%, 70%, 80%, and 90% levels of noise averages and general average
values. For 40%, 50%, and 60% levels noise, 4rd, 4rd, and 2rd were obtained the best
results, respectively (Table 11). NVBMF obtained the best results in the comparisons for

**Table 1 Compare of PSNR results for proposed method and others on six images selected from different datasets.**

| | Images | Filters | 10% | 20% | 30% | 40% | 50% | 60% | 70% | 80% | 90% |
|---|---|---|---|---|---|---|---|---|---|---|---|
| PSNR RESULTS | Lena 512 × 512 | AFMF | 38.26 | 36.77 | 35.24 | 33.76 | 32.18 | 30.83 | 28.21 | 27.44 | 22.09 |
| | | ARmF | 43.05 | 39.74 | 37.58 | 35.82 | 34.11 | 32.43 | 30.44 | 28.93 | 26.34 |
| | | IAWMF | 42.92 | 39.71 | 37.57 | 35.83 | 34.16 | 32.64 | 30.85 | 29.34 | 26.89 |
| | | ASWMF | 37.12 | 34.03 | 32.12 | 30.69 | 29.31 | 28.05 | 25.15 | 24.96 | 22.08 |
| | | AWMF | 39.01 | 37.39 | 36.14 | 34.94 | 33.55 | 32.12 | 30.18 | 28.84 | 26.31 |
| | | DAMF | 42.98 | 39.18 | 36.78 | 34.90 | 33.20 | 31.71 | 29.65 | 28.59 | 25.96 |
| | | IMF | 42.42 | 39.11 | 36.98 | 35.41 | 33.94 | 32.55 | 30.72 | 29.69 | 27.47 |
| | | TSF | 42.98 | 39.18 | 36.78 | 34.90 | 33.20 | 31.71 | 29.60 | 28.62 | 26.24 |
| | | MSCF-1 | 42.51 | 39.26 | 37.15 | 35.48 | 33.84 | 32.25 | 30.70 | 29.06 | 26.95 |
| | | DAMRmF | 43.05 | 39.73 | 37.57 | 35.83 | 34.18 | 32.68 | 31.17 | 29.39 | 27.00 |
| | | SFT_lp | 43.38 | 39.87 | 37.60 | 35.82 | 34.13 | 32.61 | 31.13 | 29.36 | 26.81 |
| | | Bilal's method | 42.47 | 38.29 | 35.63 | 33.45 | 31.48 | 28.73 | 27.92 | 25.99 | 23.03 |
| | | NVBMF | 43.61 | 39.88 | 37.32 | 35.24 | 34.30 | 33.01 | 31.68 | 30.05 | 27.59 |
| | Cameraman 512 × 512 | AFMF | 40.18 | 37.61 | 35.23 | 33.66 | 31.84 | 30.10 | 28.21 | 26.26 | 21.03 |
| | | ARmF | 44.45 | 41.14 | 38.67 | 36.77 | 34.75 | 32.62 | 30.44 | 28.34 | 25.18 |
| | | IAWMF | 44.03 | 41.01 | 38.65 | 36.72 | 34.83 | 32.88 | 30.85 | 28.82 | 25.72 |
| | | ASWMF | 35.48 | 32.60 | 30.58 | 29.18 | 27.88 | 26.63 | 25.15 | 23.34 | 20.55 |
| | | AWMF | 38.21 | 37.30 | 36.25 | 35.28 | 33.80 | 32.12 | 30.18 | 28.21 | 25.12 |
| | | DAMF | 44.12 | 40.14 | 37.06 | 35.05 | 33.15 | 31.40 | 29.65 | 27.84 | 24.83 |
| | | IMF | 43.45 | 40.11 | 37.69 | 35.93 | 34.30 | 32.46 | 30.72 | 28.85 | 26.15 |
| | | TSF | 44.12 | 40.14 | 37.06 | 35.05 | 33.16 | 31.38 | 29.60 | 27.80 | 24.90 |
| | | MSCF-1 | 43.57 | 40.36 | 37.99 | 36.26 | 34.34 | 32.35 | 30.35 | 28.47 | 25.78 |
| | | DAMRmF | 44.43 | 41.11 | 38.59 | 36.72 | 34.75 | 32.86 | 30.80 | 28.80 | 25.77 |
| | | SFT_lp | 45.36 | 41.39 | 38.69 | 36.54 | 34.50 | 32.49 | 30.17 | 28.06 | 25.13 |
| | | Bilal's method | 43.46 | 38.73 | 35.47 | 33.11 | 30.93 | 28.74 | 26.83 | 24.48 | 21.66 |
| | | NVBMF | 45.50 | 41.33 | 38.35 | 35.90 | 34.94 | 33.36 | 31.44 | 29.57 | 26.43 |
| | Airplane 512 × 512 | AFMF | 34.52 | 32.91 | 31.92 | 30.81 | 29.45 | 28.01 | 26.59 | 24.95 | 20.32 |
| | | ARmF | 41.15 | 37.73 | 35.41 | 33.59 | 31.87 | 30.01 | 28.31 | 26.56 | 23.92 |
| | | IAWMF | 40.98 | 37.66 | 35.39 | 33.66 | 31.99 | 30.21 | 28.65 | 27.02 | 24.45 |
| | | ASWMF | 34.74 | 31.70 | 29.79 | 28.33 | 26.99 | 25.65 | 24.17 | 22.52 | 19.83 |
| | | AWMF | 35.65 | 34.32 | 33.30 | 32.25 | 30.94 | 29.47 | 27.97 | 26.40 | 23.87 |
| | | DAMF | 41.27 | 36.83 | 34.42 | 32.63 | 30.88 | 29.16 | 27.70 | 26.19 | 23.59 |
| | | IMF | 40.30 | 36.96 | 34.66 | 33.05 | 31.61 | 30.02 | 28.71 | 27.22 | 24.92 |
| | | TSF | 41.27 | 36.83 | 34.42 | 32.62 | 30.87 | 29.13 | 27.64 | 26.14 | 23.71 |
| | | MSCF-1 | 39.89 | 36.60 | 34,45 | 32.83 | 31.26 | 29.62 | 28.10 | 26.63 | 24.43 |
| | | DAMRmF | 41.14 | 37.72 | 35.43 | 33.68 | 32.06 | 30.29 | 28.68 | 27.11 | 24.55 |
| | | SFT_lp | 41.89 | 38.24 | 35.53 | 33.57 | 31.83 | 30.06 | 28.43 | 26.67 | 24.11 |
| | | Bilal's method | 40.40 | 36.14 | 33.37 | 31.10 | 29.00 | 27.11 | 25.20 | 23.16 | 20.71 |
| | | NVBMF | 42.24 | 38.24 | 35.51 | 33.35 | 31.83 | 30.36 | 29.06 | 27.57 | 25.01 |

| Images | Filters | 10% | 20% | 30% | 40% | 50% | 60% | 70% | 80% | 90% |
|---|---|---|---|---|---|---|---|---|---|---|
| micromarket 2,336 × 3,504 MATLAB Library (R2020b) | AFMF | 37.67 | 35.74 | 34.03 | 32.50 | 31.04 | 29.59 | 28.10 | 26.39 | 21.15 |
| | ARmF | 43.41 | 40.05 | 37.80 | 35.90 | 34.17 | 32.37 | 30.48 | 28.43 | 25.60 |
| | IAWMF | 43.26 | 40.06 | 37.85 | 35.96 | 34.28 | 32.59 | 30.84 | 28.88 | 26.16 |
| | ASWMF | 36.38 | 32.27 | 31.33 | 29.81 | 28.47 | 27.10 | 25.55 | 23.51 | 20.44 |
| | AWMF | 38.36 | 36.94 | 35.79 | 34.61 | 33.37 | 31.88 | 30.20 | 28.27 | 25.53 |
| | DAMF | 43.19 | 39.09 | 36.38 | 34.26 | 32.60 | 31.08 | 29.58 | 27.82 | 25.14 |
| | IMF | 42.49 | 39.09 | 36.86 | 35.07 | 33.50 | 31.98 | 30.41 | 28.72 | 26.54 |
| | TSF | 43.39 | 39.28 | 36.51 | 34.35 | 32.69 | 31.17 | 29.67 | 27.93 | 25.37 |
| | MSCF-1 | 42.61 | 39.33 | 37.18 | 35.38 | 33.76 | 32.08 | 30..35 | 28.57 | 26.21 |
| | DAMRmF | 42.14 | 38.96 | 37.04 | 35.38 | 33.98 | 32.46 | 30.77 | 28.91 | 26.27 |
| | SFT_lp | 44.58 | 40.86 | 38.26 | 36.18 | 34.37 | 32.63 | 30.81 | 28.79 | 26.09 |
| | Bilal's method | 42.95 | 38.41 | 35.28 | 32.83 | 30.76 | 28.79 | 26.78 | 24.57 | 21.75 |
| | NVBMF | 44.40 | 40.48 | 37.72 | 35.45 | 34.28 | 32.88 | 31.37 | 29.60 | 26.91 |
| img_600 × 600_1 × 8 bit_B01C00GRAY_apples 600 × 600 TESTIMAGES dataset | AFMF | 38.45 | 35.65 | 33.50 | 32.09 | 31.63 | 30.86 | 29.68 | 28.07 | 21.88 |
| | ARmF | 49.76 | 46.31 | 43.91 | 41.93 | 39.89 | 38.10 | 36.17 | 34.00 | 30.83 |
| | IAWMF | 49.49 | 46.19 | 43.88 | 41.90 | 39.91 | 38.29 | 36.61 | 34.54 | 31.52 |
| | ASWMF | 42.48 | 39.10 | 37.01 | 35.13 | 33.45 | 31.65 | 29.49 | 26.40 | 21.67 |
| | AWMF | 44.75 | 43.37 | 42.16 | 40.81 | 39.24 | 37.71 | 35.96 | 33.91 | 30.78 |
| | DAMF | 49.37 | 45.55 | 42.81 | 40.77 | 38.74 | 37.18 | 35.42 | 32.88 | 28.95 |
| | IMF | 49.08 | 45.59 | 43.36 | 41.51 | 39.81 | 38.43 | 36.86 | 35.02 | 32.50 |
| | TSF | 49.36 | 45.55 | 42.81 | 40.77 | 38.74 | 37.20 | 35.57 | 33.69 | 30.39 |
| | MSCF-1 | 49.12 | 45.79 | 43.42 | 41.53 | 39.60 | 37.89 | 36.12 | 34.30 | 31.69 |
| | DAMRmF | 43.57 | 41.03 | 40.04 | 39.23 | 38.94 | 38.26 | 36.67 | 34.63 | 31.66 |
| | SFT_lp | 50.73 | 46.96 | 44.57 | 42.48 | 40.64 | 38.97 | 37.17 | 35.01 | 31.83 |
| | Bilal's method | 48.89 | 44.25 | 40.92 | 38.35 | 35.93 | 33.82 | 31.60 | 28.76 | 24.37 |
| | NVBMF | 50.41 | 46.27 | 43.46 | 41.10 | 40.48 | 39.14 | 37.66 | 35.70 | 32.59 |
| 187083 321 × 481 Berkeley 200 dataset | AFMF | 37.67 | 36.30 | 34.91 | 33.66 | 32.22 | 30.72 | 29.26 | 27.62 | 21.31 |
| | ARmF | 42.85 | 39.46 | 37.40 | 35.80 | 34.12 | 32.58 | 30.90 | 28.96 | 26.71 |
| | IAWMF | 42.86 | 39.53 | 37.50 | 35.87 | 34.25 | 32.76 | 31.20 | 29.37 | 27.20 |
| | ASWMF | 37.61 | 34.45 | 32.57 | 31.14 | 29.87 | 28.61 | 27.19 | 25.29 | 22.43 |
| | AWMF | 38.94 | 37.14 | 35.83 | 34.75 | 33.49 | 32.18 | 30.68 | 28.84 | 26.67 |
| | DAMF | 43.05 | 39.11 | 36.54 | 34.86 | 33.19 | 31.76 | 30.29 | 28.60 | 26.38 |
| | IMF | 42.03 | 38.72 | 36.67 | 35.18 | 33.77 | 32.45 | 31.25 | 29.68 | 27.82 |
| | TSF | 43.05 | 39.11 | 36.54 | 34.86 | 33.20 | 31.74 | 30.32 | 28.66 | 26.66 |
| | MSCF-1 | 42.14 | 38.84 | 36.82 | 35.35 | 33.74 | 32.32 | 30.80 | 29.10 | 27.22 |
| | DAMRmF | 42.84 | 39.45 | 37.39 | 35.78 | 34.20 | 32.73 | 31.22 | 29.45 | 27.34 |
| | SFT_lp | 43.84 | 39.93 | 37.74 | 35.90 | 34.22 | 32.70 | 31.26 | 29.47 | 27.30 |
| | Bilal's method | 43.08 | 38.40 | 35.68 | 33.77 | 31.85 | 30.08 | 28.37 | 26.42 | 23.99 |
| | NVBMF | 43.75 | 39.87 | 37.53 | 35.54 | 34.12 | 32.86 | 31.58 | 29.92 | 27.78 |

**Table 2 Compare of SSIM results for proposed method and others on six images selected from different datasets.**

|  | Images | Filters | 10% | 20% | 30% | 40% | 50% | 60% | 70% | 80% | 90% |
|---|---|---|---|---|---|---|---|---|---|---|---|
| SSIM RESULTS | Lena 512 × 512 | AFMF | 0.9614 | 0.9580 | .0.9499 | 0.9375 | 0.9191 | 0.8962 | 0.8634 | 0.8155 | 0.6866 |
|  |  | ARmF | 0.9912 | 0.9812 | 0.9696 | 0.9561 | 0.9393 | 0.9177 | 0.8881 | 0.8458 | 0.7744 |
|  |  | IAWMF | 0.9911 | 0.9812 | 0.9695 | 0.9561 | 0.9397 | 0.9200 | 0.8936 | 0.8555 | 0.7916 |
|  |  | ASWMF | 0.9787 | 0.9557 | 0.9314 | 0.9044 | 0.8734 | 0.8375 | 0.7903 | 0.7302 | 0.6355 |
|  |  | AWMF | 0.9822 | 0.9737 | 0.9634 | 0.9507 | 0.9344 | 0.9136 | 0.8848 | 0.8434 | 0.7730 |
|  |  | DAMF | 0.9904 | 0.9790 | 0.9655 | 0.9497 | 0.9308 | 0.9086 | 0.8794 | 0.8384 | 0.7654 |
|  |  | IMF | 0.9904 | 0.9798 | 0.9676 | 0.9545 | 0.9392 | 0.9206 | 0.8974 | 0.8636 | 0.8084 |
|  |  | TSF | 0.9904 | 0.9790 | 0.9655 | 0.9497 | 0.9308 | 0.9087 | 0.8796 | 0.8398 | 0.7723 |
|  |  | MSCF-1 | 0.9905 | 0.9800 | 0.9678 | 0.9539 | 0.9369 | 0.9154 | 0.8865 | 0.8487 | 0.7924 |
|  |  | DAMRmF | 0.9912 | 0.9812 | 0.9696 | 0.9565 | 0.9405 | 0.9212 | 0.8952 | 0.8581 | 0.7958 |
|  |  | SFT_lp | 0.9914 | 0.9817 | 0.9702 | 0.9573 | 0.9415 | 0.9223 | 0.8965 | 0.8587 | 0.7935 |
|  |  | Bilal's Method | 0.9899 | 0.9765 | 0.9602 | 0.9397 | 0.9145 | 0.8829 | 0.8414 | 0.7875 | 0.6984 |
|  |  | NVBMF | 0.9915 | 0.9811 | 0.9676 | 0.9513 | 0.9416 | 0.9247 | 0.9025 | 0.8698 | 0.8120 |
|  | Cameraman 512 × 512 | AFMF | 0.9892 | 0.9850 | 0.9788 | 0.9705 | 0.9572 | 0.9393 | 0.9127 | 0.8693 | 0.7390 |
|  |  | ARmF | 0.9972 | 0.9938 | 0.9896 | 0.9839 | 0.9752 | 0.9630 | 0.9439 | 0.9122 | 0.8459 |
|  |  | IAWMF | 0.9969 | 0.9937 | 0.9896 | 0.9839 | 0.9757 | 0.9646 | 0.9474 | 0.9186 | 0.8575 |
|  |  | ASWMF | 0.9800 | 0.9600 | 0.9363 | 0.9110 | 0.8789 | 0.8406 | 0.7886 | 0.7158 | 0.6050 |
|  |  | AWMF | 0.9884 | 0.9857 | 0.9821 | 0.9772 | 0.9693 | 0.9580 | 0.9398 | 0.9089 | 0.8434 |
|  |  | DAMF | 0.9967 | 0.9921 | 0.9859 | 0.9781 | 0.9673 | 0.9541 | 0.9355 | 0.9046 | 0.8575 |
|  |  | IMF | 0.9964 | 0.9921 | 0.9864 | 0.9793 | 0.9696 | 0.9558 | 0.9381 | 0.9079 | 0.8496 |
|  |  | TSF | 0.9967 | 0.9921 | 0.9859 | 0.9781 | 0.9673 | 0.9540 | 0.9351 | 0.9045 | 0.8393 |
|  |  | MSCF-1 | 0.9966 | 0.9928 | 0.9880 | 0.9820 | 0.9728 | 0.9606 | 0.9419 | 0.9130 | 0.8567 |
|  |  | DAMRmF | 0.9971 | 0.9938 | 0.9895 | 0.9836 | 0.9748 | 0.9634 | 0.9457 | 0.9168 | 0.8563 |
|  |  | SFT_lp | 0.9972 | 0.9935 | 0.9884 | 0.9812 | 0.9706 | 0.9563 | 0.9333 | 0.8975 | 0.8279 |
|  |  | Bilal's method | 0.9961 | 0.9894 | 0.9798 | 0.9665 | 0.9464 | 0.9193 | 0.8829 | 0.8285 | 0.7430 |
|  |  | NVBMF | 0.9978 | 0.9943 | 0.9895 | 0.9820 | 0.9736 | 0.9630 | 0.9480 | 0.9226 | 0.8664 |
|  | Airplane 512 × 512 | AFMF | 0.9731 | 0.9667 | 0.9599 | 0.9488 | 0.9321 | 0.9094 | 0.8768 | 0.8301 | 0.6990 |
|  |  | ARmF | 0.9939 | 0.9868 | 0.9786 | 0.9679 | 0.9541 | 0.9346 | 0.9070 | 0.8682 | 0.7951 |
|  |  | IAWMF | 0.9938 | 0.9868 | 0.9787 | 0.9683 | 0.9550 | 0.9368 | 0.9122 | 0.8771 | 0.8107 |
|  |  | ASWMF | 0.9774 | 0.9536 | 0.9276 | 0.8978 | 0.8621 | 0.8166 | 0.7571 | 0.6782 | 0.5561 |
|  |  | AWMF | 0.9834 | 0.9766 | 0.9693 | 0.9595 | 0.9463 | 0.9278 | 0.9011 | 0.8636 | 0.7922 |
|  |  | DAMF | 0.9932 | 0.9842 | 0.9741 | 0.9614 | 0.9455 | 0.9246 | 0.8977 | 0.8604 | 0.7858 |
|  |  | IMF | 0.9929 | 0.9847 | 0.9752 | 0.9638 | 0.9502 | 0.9310 | 0.9083 | 0.8745 | 0.8127 |
|  |  | TSF | 0.9932 | 0.9842 | 0.9741 | 0.9614 | 0.9454 | 0.9244 | 0.8970 | 0.8601 | 0.7910 |
|  |  | MSCF-1 | 0.9928 | 0.9846 | 0.9754 | 0.9641 | 0.9497 | 0.9301 | 0,9034 | 0.8686 | 0.8084 |
|  |  | DAMRmF | 0.9939 | 0.9867 | 0.9785 | 0.9681 | 0.9550 | 0.9369 | 0.9118 | 0.8779 | 0.8120 |
|  |  | SFT_lp | 0.9942 | 0.9870 | 0.9781 | 0.9666 | 0.9520 | 0.9320 | 0.9048 | 0.8642 | 0.7915 |
|  |  | Bilal's Method | 0.9924 | 0.9815 | 0.9677 | 0.9485 | 0.9233 | 0.8890 | 0.8425 | 0.7814 | 0.6848 |
|  |  | NVBMF | 0.9948 | 0.9877 | 0.9787 | 0.9659 | 0.9532 | 0.9369 | 0.9159 | 0.8852 | 0.8239 |

| Images | Filters | 10% | 20% | 30% | 40% | 50% | 60% | 70% | 80% | 90% |
|---|---|---|---|---|---|---|---|---|---|---|
| micromarket 2,336 × 3,504 MATLAB Library (R2020b) | AFMF | 0.9813 | 0.9766 | 0.9693 | 0.9586 | 0.9439 | 0.9235 | 0.8941 | 0.8484 | 0.7147 |
| | ARmF | 0.9957 | 0.9908 | 0.9848 | 0.9770 | 0.9664 | 0.9512 | 0.9284 | 0.8923 | 0.8202 |
| | IAWMF | 0.9956 | 0.9908 | 0.9849 | 0.9772 | 0.9671 | 0.9532 | 0.9329 | 0.9005 | 0.8362 |
| | ASWMF | 0.9798 | 0.9584 | 0.9347 | 0.9071 | 0.8739 | 0.8315 | 0.7747 | 0.6922 | 0.5664 |
| | AWMF | 0.9868 | 0.9822 | 0.9769 | 0.9698 | 0.9602 | 0.9457 | 0.9237 | 0.8883 | 0.8172 |
| | DAMF | 0.9952 | 0.9888 | 0.9803 | 0.9694 | 0.9563 | 0.9397 | 0.9173 | 0.8823 | 0.8099 |
| | IMF | 0.9948 | 0.9888 | 0.9815 | 0.9724 | 0.9607 | 0.9452 | 0.9233 | 0.8911 | 0.8359 |
| | TSF | 0.9953 | 0.9889 | 0.9804 | 0.9695 | 0.9564 | 0.9399 | 0.9176 | 0.8833 | 0.8138 |
| | MSCF-1 | 0. 9950 | 0.9894 | 0.9828 | 0.9744 | 0.9635 | 0.9482 | 0.9261 | 0.8941 | 0.8362 |
| | DAMRmF | 0.9952 | 0.9901 | 0.9842 | 0.9763 | 0.9663 | 0.9523 | 0.9319 | 0.9002 | 0.8380 |
| | SFT_lp | 0.9665 | 0.9919 | 0.9859 | 0.9779 | 0.9671 | 0.9522 | 0.9299 | 0.8944 | 0.8273 |
| | Bilal's method | 0.9950 | 0.9868 | 0.9749 | 0.9581 | 0.9354 | 0.9044 | 0.8602 | 0.7957 | 0.6938 |
| | NVBMF | 0.9964 | 0.9915 | 0.9846 | 0.9749 | 0.9661 | 0.9540 | 0.9369 | 0.9099 | 0.8533 |
| img_600 × 600_1 × 8 bit_B01C00GRAY_apples 600 × 600 TESTIMAGES dataset | AFMF | 0.9857 | 0.9814 | 0.9762 | 0.9687 | 0.9606 | 0.9488 | 0.9291 | 0.8982 | 0.7751 |
| | ARmF | 0.9973 | 0.9943 | 0.9905 | 0.9856 | 0.9791 | 0.9695 | 0.9547 | 0.9314 | 0.8840 |
| | IAWMF | 0.9972 | 0.9943 | 0.9905 | 0.9856 | 0.9794 | 0.9707 | 0.9580 | 0.9375 | 0.8970 |
| | ASWMF | 0.9874 | 0.9734 | 0.9569 | 0.9364 | 0.9109 | 0.8769 | 0.8275 | 0.7526 | 0.6316 |
| | AWMF | 0.9924 | 0.9897 | 0.9864 | 0.9818 | 0.9757 | 0.9667 | 0.9523 | 0.9296 | 0.8828 |
| | DAMF | 0.9967 | 0.9928 | 0.9877 | 0.9813 | 0.9736 | 0.9635 | 0.9484 | 0.9242 | 0.8726 |
| | IMF | 0.9969 | 0.9934 | 0.9892 | 0.9839 | 0.9775 | 0.9696 | 0.9577 | 0.9405 | 0.9110 |
| | TSF | 0.9967 | 0.9928 | 0.9877 | 0.9813 | 0.9735 | 0.9634 | 0.9487 | 0.9263 | 0.8797 |
| | MSCF-1 | 0.9970 | 0.9936 | 0.9895 | 0.9843 | 0.9776 | 0.9680 | 0.9538 | 0.9340 | 0.8986 |
| | DAMRmF | 0.9957 | 0.9920 | 0.9885 | 0.9839 | 0.9787 | 0.9707 | 0.9582 | 0.9385 | 0.9004 |
| | SFT_lp | 0.9973 | 0.9942 | 0.9904 | 0.9852 | 0.9789 | 0.9702 | 0.9571 | 0.9373 | 0.9004 |
| | Bilal's method | 0.9962 | 0.9909 | 0.9834 | 0.9726 | 0.9584 | 0.9396 | 0.9122 | 0.8697 | 0.7899 |
| | NVBMF | 0.9976 | 0.9945 | 0.9900 | 0.9837 | 0.9801 | 0.9734 | 0.9635 | 0.9478 | 0.9137 |
| 187083 321 × 481 Berkeley 200 sataset | AFMF | 0.9766 | 0.9702 | 0.9624 | 0.9504 | 0.9343 | 0.9095 | 0.8771 | 0.8287 | 0.6773 |
| | ARmF | 0.9942 | 0.9869 | 0.9789 | 0.9686 | 0.9555 | 0.9358 | 0.9089 | 0.8684 | 0.7941 |
| | IAWMF | 0.9942 | 0.9871 | 0.9793 | 0.9692 | 0.9565 | 0.9384 | 0.9143 | 0.8784 | 0.8130 |
| | ASWMF | 0.9807 | 0.9588 | 0.9361 | 0.9086 | 0.8776 | 0.83.72 | 0.7828 | 0.7071 | 0.5958 |
| | AWMF | 0.9859 | 0.9786 | 0.9709 | 0.9614 | 0.9491 | 0.9299 | 0.9039 | 0.8645 | 0.7917 |
| | DAMF | 0.9937 | 0.9849 | 0.9743 | 0.9615 | 0.9459 | 0.9246 | 0.8979 | 0.8588 | 0.7845 |
| | IMF | 0.9931 | 0.9849 | 0.9759 | 0.9652 | 0.9525 | 0.9351 | 0.9143 | 0.8824 | 0.8307 |
| | TSF | 0.9937 | 0.9849 | 0.9743 | 0.9615 | 0.9459 | 0.9245 | 0.8986 | 0.8603 | 0.7906 |
| | MSCF-1 | 0.9933 | 0.9851 | 0.9764 | 0.9655 | 0.9520 | 0.9323 | 0.9067 | 0.8713 | 0.8135 |
| | DAMRmF | 0.9942 | 0.9869 | 0.9789 | 0.9687 | 0.9561 | 0.9381 | 0.9143 | 0.8794 | 0.8168 |
| | SFT_lp | 0.9951 | 0.9880 | 0.9805 | 0.9699 | 0.9569 | 0.9389 | 0.9147 | 0.8763 | 0.8097 |
| | Bilal's method | 0.9936 | 0.9825 | 0.9687 | 0.9505 | 0.9260 | 0.8926 | 0.8472 | 0.7796 | 0.6755 |
| | NVBMF | 0.9951 | 0.9880 | 0.9789 | 0.9667 | 0.9558 | 0.9403 | 0.9204 | 0.8904 | 0.8332 |

**Table 3 Compare of IEF results for proposed method and others on six images selected from different datasets.**

| | Images | Filters | 10% | 20% | 30% | 40% | 50% | 60% | 70% | 80% | 90% |
|---|---|---|---|---|---|---|---|---|---|---|---|
| IEF RESULTS | Lena 512 × 512 | AFMF | 186.43 | 251.76 | 241.80 | 226.42 | 196.76 | 175.00 | 136.45 | 109.51 | 39.23 |
| | | ARmF | 631.88 | 608.86 | 524.67 | 471.33 | 398.16 | 326.41 | 245.77 | 188.22 | 115.22 |
| | | IAWMF | 613.85 | 604.65 | 522.92 | 470.52 | 403.32 | 343.27 | 265.65 | 207.72 | 132.74 |
| | | ASWMF | 160.46 | 161.99 | 150.37 | 142.41 | 131.32 | 116.96 | 92.68 | 70.30 | 38.69 |
| | | AWMF | 253.68 | 359.82 | 380.11 | 384.37 | 349.40 | 301.78 | 234.21 | 183.52 | 113.79 |
| | | DAMF | 650.04 | 545.25 | 443.73 | 382.47 | 321.61 | 274.61 | 217.59 | 173.06 | 105.92 |
| | | IMF | 545.62 | 530.80 | 464.30 | 435.60 | 397.75 | 351.27 | 284.99 | 233.85 | 156.12 |
| | | TSF | 650.04 | 545.25 | 443.73 | 382.64 | 322.00 | 275.07 | 219.49 | 177.08 | 116.23 |
| | | MSCF-1 | 557.16 | 549.21 | 476.87 | 435.16 | 374.48 | 314.36 | 241.16 | 195.17 | 135.02 |
| | | DAMRmF | 631.47 | 606.74 | 522.76 | 473.67 | 409.04 | 351.36 | 270.76 | 212.62 | 135.55 |
| | | SFT_lp | 718.62 | 671.18 | 565.12 | 510.21 | 448.88 | 388.06 | 296.98 | 237.06 | 145.41 |
| | | Bilal's method | 540.19 | 417.47 | 317.83 | 255.54 | 208.82 | 162.77 | 119.27 | 87.04 | 46.24 |
| | | NVBMF | 700.97 | 608.04 | 522.92 | 397.87 | 435.15 | 393.25 | 316.14 | 254.35 | 159.91 |
| | Cameraman 512 × 512 | AFMF | 269.42 | 301.37 | 261.09 | 239.54 | 197.60 | 156.83 | 129.51 | 96.39 | 33.80 |
| | | ARmF | 917.51 | 943.94 | 780.82 | 663.01 | 506.12 | 356.88 | 271.62 | 194.25 | 106.41 |
| | | IAWMF | 839.58 | 917.45 | 782.49 | 655.63 | 520.60 | 377.71 | 303.39 | 219.38 | 122.36 |
| | | ASWMF | 122.53 | 132.85 | 123.13 | 117.49 | 107.06 | 90.68 | 77.02 | 53.74 | 28.32 |
| | | AWMF | 234.15 | 394.22 | 451.87 | 470.92 | 411.27 | 319.98 | 255.46 | 188.17 | 104.80 |
| | | DAMF | 856.57 | 727.56 | 543.78 | 444.62 | 352.05 | 269.87 | 228.41 | 174.07 | 98.02 |
| | | IMF | 749.08 | 758.36 | 642.09 | 574.57 | 486.92 | 366.83 | 308.68 | 230.37 | 140.90 |
| | | TSF | 856.57 | 727.56 | 543.81 | 444.96 | 351.57 | 269.90 | 231.48 | 177.21 | 103.61 |
| | | MSCF-1 | 753.39 | 791.27 | 664.28 | 590.61 | 464.57 | 337.18 | 268.95 | 201.55 | 123.25 |
| | | DAMRmF | 916.33 | 933.92 | 771.81 | 662.49 | 514.55 | 379.96 | 302.87 | 221.25 | 125.73 |
| | | SFT_lp | 1,249.40 | 1,138.00 | 902.27 | 734.14 | 555.08 | 403.81 | 304.84 | 213.71 | 128.71 |
| | | Bilal's Method | 743.11 | 528.60 | 364.15 | 274.21 | 198.75 | 140.55 | 110.20 | 74.29 | 39.56 |
| | | NVBMF | 1,127.50 | 950.65 | 710.67 | 536.78 | 564.89 | 448.84 | 364.57 | 271.12 | 147.84 |
| | Airplane 512 × 512 | AFMF | 22.15 | 30.86 | 37.35 | 40.31 | 37.01 | 33.46 | 29.07 | 23.17 | 12.87 |
| | | ARmF | 154.02 | 129.55 | 110.24 | 99.11 | 84.67 | 66.09 | 54.58 | 41.53 | 24.63 |
| | | IAWMF | 141.41 | 123.84 | 107.38 | 99.48 | 86.40 | 68.98 | 58.64 | 45.50 | 27.29 |
| | | ASWMF | 34.70 | 33.61 | 31.87 | 31.70 | 29.12 | 26.30 | 23.00 | 18.63 | 13.30 |
| | | AWMF | 35.12 | 52.04 | 63.25 | 69.96 | 65.24 | 56.95 | 49.45 | 39.38 | 24.20 |
| | | DAMF | 140.58 | 98.04 | 83.33 | 79.17 | 67.25 | 55.10 | 47.75 | 38.12 | 23.32 |
| | | IMF | 126.41 | 107.55 | 92.75 | 87.13 | 78.32 | 63.97 | 56.93 | 45.76 | 29.31 |
| | | TSF | 140.58 | 98.04 | 83.33 | 79.07 | 67.02 | 54.60 | 46.47 | 36.77 | 23.47 |
| | | MSCF-1 | 106.08 | 93.42 | 84.11 | 80.31 | 71.01 | 58.96 | 50.91 | 41.53 | 27.07 |
| | | DAMRmF | 153.08 | 128.96 | 111.97 | 102.39 | 89.38 | 71.30 | 59.57 | 46.63 | 28.00 |
| | | SFT_lp | 171.61 | 136.24 | 105.63 | 91.64 | 77.39 | 60.57 | 50.25 | 38.25 | 23.29 |
| | | Bilal's Method | 112.25 | 81.07 | 65.21 | 52.84 | 41.05 | 32.75 | 24.61 | 17.88 | 10.82 |
| | | NVBMF | 192.83 | 145.50 | 113.20 | 94.40 | 79.85 | 67.68 | 61.45 | 49.82 | 30.40 |

| Table 3 (continued) | | | | | | | | | | |
|---|---|---|---|---|---|---|---|---|---|---|
| Images | Filters | 10% | 20% | 30% | 40% | 50% | 60% | 70% | 80% | 90% |
| micromarket 2,336 × 3,504 MATLAB Library (R2020b) | AFMF | 284.20 | 332.03 | 318.87 | 287.72 | 250.51 | 213.41 | 177.32 | 138.50 | 43.48 |
| | ARmF | 1,051.90 | 956.73 | 844.21 | 722.14 | 598.47 | 468.10 | 349.22 | 252.11 | 145.65 |
| | IAWMF | 1,020.40 | 960.57 | 853.00 | 731.14 | 615.16 | 494.33 | 382.49 | 282.70 | 168.43 |
| | ASWMF | 216.40 | 207.76 | 196.60 | 183.03 | 164.23 | 140.12 | 110.39 | 74.97 | 38.29 |
| | AWMF | 337.59 | 473.15 | 533.58 | 537.74 | 495.05 | 416.82 | 326.04 | 242.39 | 143.11 |
| | DAMF | 1,155.20 | 855.36 | 647.65 | 516.65 | 429.13 | 357.68 | 291.07 | 224.22 | 133.42 |
| | IMF | 843.49 | 770.50 | 690.94 | 614.87 | 535.93 | 453.81 | 370.45 | 292.57 | 197.73 |
| | TSF | 1,155.20 | 855.36 | 647.65 | 516.62 | 429.16 | 358.65 | 295.28 | 230.32 | 143.16 |
| | MSCF-1 | 871.04 | 807.17 | 729.51 | 640.05 | 543.30 | 438.62 | 340.58 | 263.03 | 171.29 |
| | DAMRmF | 1,051.50 | 954.69 | 840.27 | 721.91 | 611.20 | 494.25 | 384.72 | 289.53 | 175.07 |
| | SFT_lp | 1,554.10 | 1,292.50 | 1,059.70 | 879.19 | 722.02 | 579.76 | 449.80 | 332.62 | 203.02 |
| | Bilal's Method | 965.98 | 638.66 | 456.40 | 339.17 | 255.90 | 190.86 | 138.26 | 93.24 | 52.59 |
| | NVBMF | 1,297.30 | 1,031.40 | 801.71 | 629.90 | 637.96 | 552.87 | 454.50 | 350.90 | 206.72 |
| img_600 × 600_1 × 8 bit_B01C00GRAY_apples 600 × 600 TESTIMAGES dataset | AFMF | 470.95 | 502.95 | 512.86 | 471.98 | 380.29 | 334.53 | 261.97 | 195.57 | 49.48 |
| | ARmF | 1,694.90 | 1,442.00 | 1,326.50 | 1,119.90 | 883.60 | 700.11 | 518.25 | 385.82 | 213.07 |
| | IAWMF | 1,580.60 | 1,397.80 | 1,321.00 | 1,110.90 | 881.89 | 731.58 | 569.12 | 435.93 | 246.16 |
| | ASWMF | 302.49 | 268.79 | 258.25 | 215.57 | 188.87 | 147.36 | 112.09 | 67.78 | 30.73 |
| | AWMF | 505.16 | 721.97 | 857.11 | 830.39 | 745.28 | 631.68 | 485.98 | 374.15 | 209.17 |
| | DAMF | 1,505.70 | 1,145.10 | 952.06 | 800.74 | 647.00 | 552.42 | 437.06 | 348.72 | 191.45 |
| | IMF | 1,423.40 | 1,203.80 | 1137.60 | 988.97 | 834.49 | 722.79 | 578.45 | 455.25 | 290.76 |
| | TSF | 1,505.60 | 1,145.10 | 952.03 | 800.13 | 645.49 | 549.88 | 437.63 | 335.30 | 200.91 |
| | MSCF-1 | 1,462.60 | 1,276.80 | 1,175.00 | 1,012.70 | 822.11 | 664.86 | 508.61 | 409.45 | 252.94 |
| | DAMRmF | 1,692.50 | 1,436.20 | 1,320.00 | 1,116.80 | 897.59 | 754.04 | 584.60 | 441.54 | 261.31 |
| | SFT_lp | 2,066.20 | 1,625.50 | 1,484.40 | 1,221.20 | 979.54 | 788.35 | 594.70 | 433.37 | 228.35 |
| | Bilal's method | 1,218.30 | 758.77 | 524.63 | 397.00 | 277.88 | 196.18 | 136.28 | 79.87 | 32.74 |
| | NVBMF | 1,900.10 | 1,387.90 | 1,173.20 | 920.19 | 1,014.60 | 902.58 | 735.32 | 575.65 | 319.11 |
| 187083 321 × 481 Berkeley 200 dataset | AFMF | 236.34 | 328.58 | 330.02 | 332.01 | 285.55 | 252.38 | 204.25 | 169.75 | 40.85 |
| | ARmF | 990.68 | 822.00 | 758.39 | 712.57 | 588.65 | 500.28 | 372.28 | 292.65 | 199.07 |
| | IAWMF | 987.55 | 835.33 | 774.81 | 725.89 | 603.88 | 519.75 | 398.53 | 324.01 | 225.00 |
| | ASWMF | 272.85 | 252.56 | 236.63 | 225.08 | 202.12 | 175.84 | 133.46 | 91.07 | 46.90 |
| | AWMF | 374.69 | 488.77 | 530.40 | 553.48 | 506.17 | 453.77 | 352.01 | 282.76 | 196.27 |
| | DAMF | 1,082.00 | 784.44 | 602.91 | 573.09 | 461.20 | 413.37 | 320.28 | 268.28 | 179.84 |
| | IMF | 813.75 | 699.01 | 651.44 | 626.73 | 551.45 | 492.96 | 415.68 | 351.28 | 261.39 |
| | TSF | 1,082.00 | 784.44 | 602.91 | 573.13 | 461.24 | 411.89 | 323.44 | 271.35 | 199.27 |
| | MSCF-1 | 839.53 | 715.23 | 668.08 | 642.69 | 536.77 | 473.27 | 364.15 | 300.23 | 224.52 |
| | DAMRmF | 989.81 | 822.61 | 757.11 | 713.34 | 598.34 | 522.67 | 403.57 | 329.16 | 233.76 |
| | SFT_lp | 1282.30 | 969.91 | 851.28 | 755.33 | 626.65 | 543.58 | 427.27 | 338.28 | 233.18 |
| | Bilal's method | 1,035.70 | 675.62 | 497.08 | 446.73 | 340.65 | 264.32 | 200.56 | 140.56 | 82.59 |
| | NVBMF | 1,197.70 | 878.73 | 756.20 | 652.12 | 597.69 | 537.22 | 445.71 | 373.30 | 258.41 |

**Table 4 Mean PSNR results for the 200 Berkeley dataset (BSDS) images with different SPN ratios.** AWMF was realized with 199 images. (AWMF was unable to process image 292066 in the Berkeley dataset).

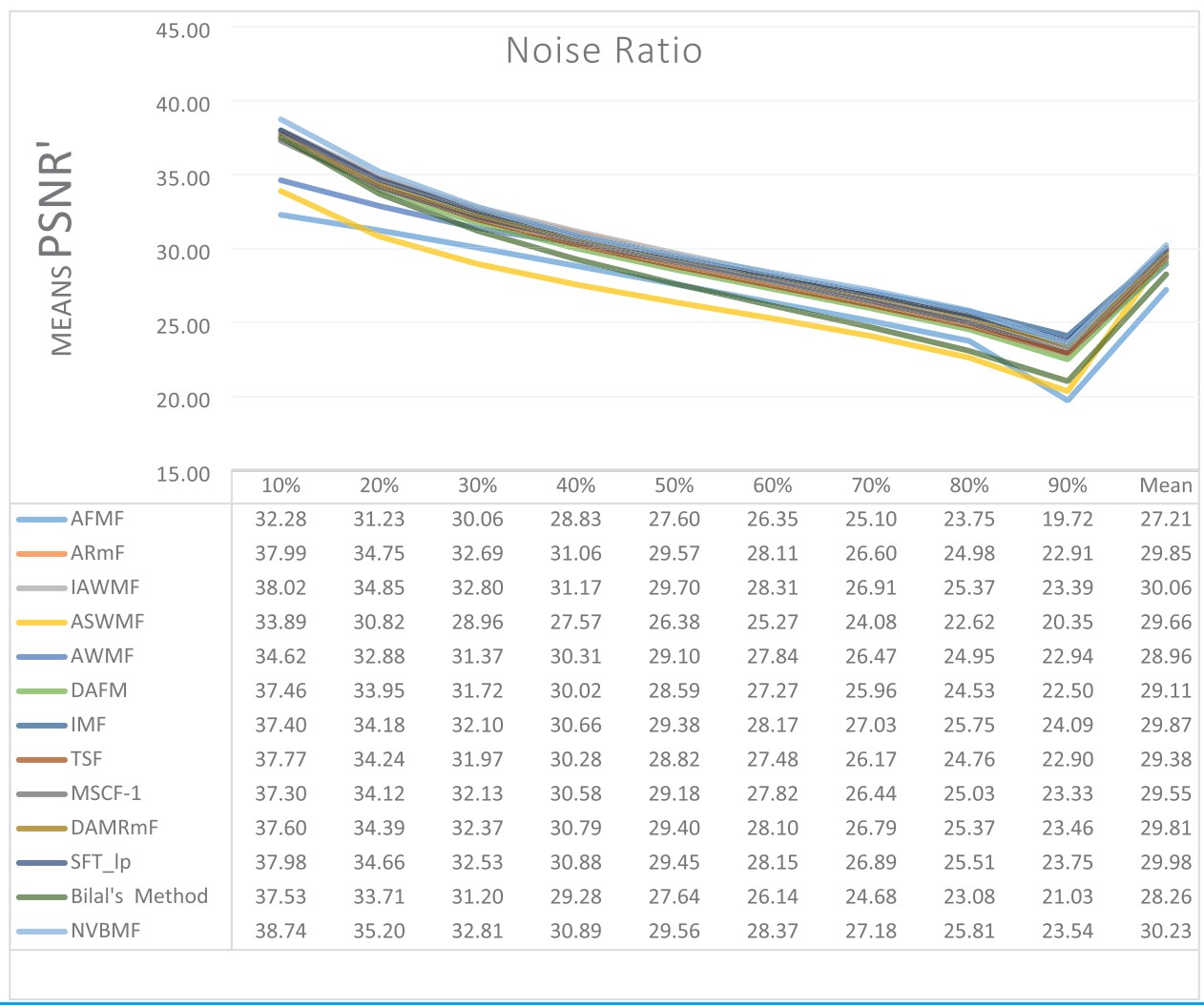

| | 10% | 20% | 30% | 40% | 50% | 60% | 70% | 80% | 90% | Mean |
|---|---|---|---|---|---|---|---|---|---|---|
| AFMF | 32.28 | 31.23 | 30.06 | 28.83 | 27.60 | 26.35 | 25.10 | 23.75 | 19.72 | 27.21 |
| ARmF | 37.99 | 34.75 | 32.69 | 31.06 | 29.57 | 28.11 | 26.60 | 24.98 | 22.91 | 29.85 |
| IAWMF | 38.02 | 34.85 | 32.80 | 31.17 | 29.70 | 28.31 | 26.91 | 25.37 | 23.39 | 30.06 |
| ASWMF | 33.89 | 30.82 | 28.96 | 27.57 | 26.38 | 25.27 | 24.08 | 22.62 | 20.35 | 29.66 |
| AWMF | 34.62 | 32.88 | 31.37 | 30.31 | 29.10 | 27.84 | 26.47 | 24.95 | 22.94 | 28.96 |
| DAFM | 37.46 | 33.95 | 31.72 | 30.02 | 28.59 | 27.27 | 25.96 | 24.53 | 22.50 | 29.11 |
| IMF | 37.40 | 34.18 | 32.10 | 30.66 | 29.38 | 28.17 | 27.03 | 25.75 | 24.09 | 29.87 |
| TSF | 37.77 | 34.24 | 31.97 | 30.28 | 28.82 | 27.48 | 26.17 | 24.76 | 22.90 | 29.38 |
| MSCF-1 | 37.30 | 34.12 | 32.13 | 30.58 | 29.18 | 27.82 | 26.44 | 25.03 | 23.33 | 29.55 |
| DAMRmF | 37.60 | 34.39 | 32.37 | 30.79 | 29.40 | 28.10 | 26.79 | 25.37 | 23.46 | 29.81 |
| SFT_lp | 37.98 | 34.66 | 32.53 | 30.88 | 29.45 | 28.15 | 26.89 | 25.51 | 23.75 | 29.98 |
| Bilal's Method | 37.53 | 33.71 | 31.20 | 29.28 | 27.64 | 26.14 | 24.68 | 23.08 | 21.03 | 28.26 |
| NVBMF | 38.74 | 35.20 | 32.81 | 30.89 | 29.56 | 28.37 | 27.18 | 25.81 | 23.54 | 30.23 |

IEF with 70%, 80%, and 90%. For 10%, 20%, 30%, 40%, 50%, 60% levels of noise and general average values, the 2rd, 2rd, 5rd, 7rd, 2rd, 2rd and 2rd were obtained the best results, respectively (Table 12).

## Statistical tests to compare performance between methods

In this section, the significance made of the comparisons between the method developed in this study and the state-of-the-art methods were tested. The significance test was performed with Wilcoxon signed-rank test because the data did not show normal distribution (*Hung et al., 2022*; *Cevahir, 2020*). The effect value of significance was determined by the Pearson Correlation Coefficient (r) (*Cevahir, 2020*). The results obtained are given in Tables 13–15.

**Table 5 Mean SSIM results for the 200 Berkeley dataset (BSDS) images with different SPN ratios.** AWMF was realized with 199 images. (AWMF was unable to process image 292066 in the Berkeley dataset).

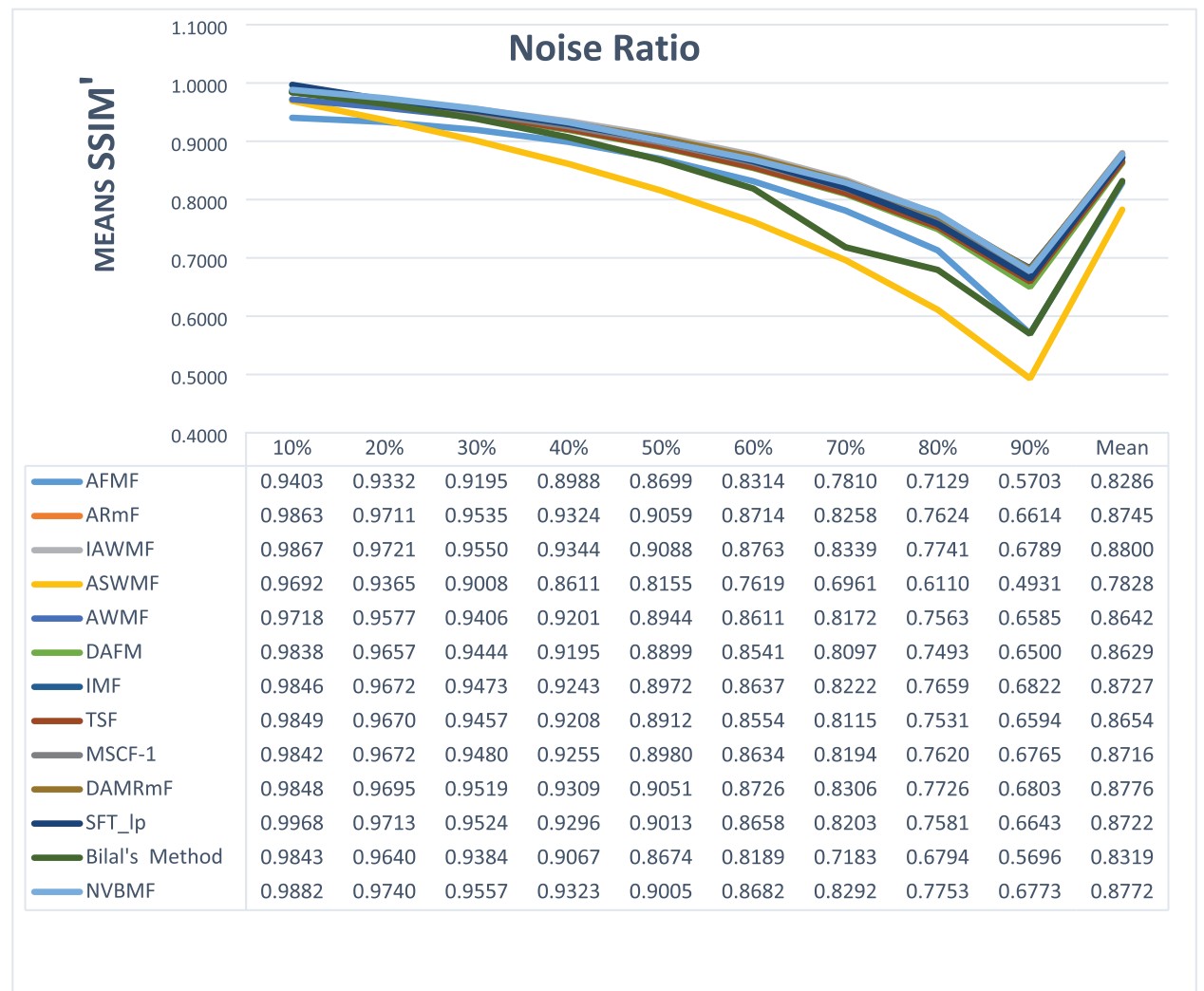

| | 10% | 20% | 30% | 40% | 50% | 60% | 70% | 80% | 90% | Mean |
|---|---|---|---|---|---|---|---|---|---|---|
| AFMF | 0.9403 | 0.9332 | 0.9195 | 0.8988 | 0.8699 | 0.8314 | 0.7810 | 0.7129 | 0.5703 | 0.8286 |
| ARmF | 0.9863 | 0.9711 | 0.9535 | 0.9324 | 0.9059 | 0.8714 | 0.8258 | 0.7624 | 0.6614 | 0.8745 |
| IAWMF | 0.9867 | 0.9721 | 0.9550 | 0.9344 | 0.9088 | 0.8763 | 0.8339 | 0.7741 | 0.6789 | 0.8800 |
| ASWMF | 0.9692 | 0.9365 | 0.9008 | 0.8611 | 0.8155 | 0.7619 | 0.6961 | 0.6110 | 0.4931 | 0.7828 |
| AWMF | 0.9718 | 0.9577 | 0.9406 | 0.9201 | 0.8944 | 0.8611 | 0.8172 | 0.7563 | 0.6585 | 0.8642 |
| DAFM | 0.9838 | 0.9657 | 0.9444 | 0.9195 | 0.8899 | 0.8541 | 0.8097 | 0.7493 | 0.6500 | 0.8629 |
| IMF | 0.9846 | 0.9672 | 0.9473 | 0.9243 | 0.8972 | 0.8637 | 0.8222 | 0.7659 | 0.6822 | 0.8727 |
| TSF | 0.9849 | 0.9670 | 0.9457 | 0.9208 | 0.8912 | 0.8554 | 0.8115 | 0.7531 | 0.6594 | 0.8654 |
| MSCF-1 | 0.9842 | 0.9672 | 0.9480 | 0.9255 | 0.8980 | 0.8634 | 0.8194 | 0.7620 | 0.6765 | 0.8716 |
| DAMRmF | 0.9848 | 0.9695 | 0.9519 | 0.9309 | 0.9051 | 0.8726 | 0.8306 | 0.7726 | 0.6803 | 0.8776 |
| SFT_lp | 0.9968 | 0.9713 | 0.9524 | 0.9296 | 0.9013 | 0.8658 | 0.8203 | 0.7581 | 0.6643 | 0.8722 |
| Bilal's Method | 0.9843 | 0.9640 | 0.9384 | 0.9067 | 0.8674 | 0.8189 | 0.7183 | 0.6794 | 0.5696 | 0.8319 |
| NVBMF | 0.9882 | 0.9740 | 0.9557 | 0.9323 | 0.9005 | 0.8682 | 0.8292 | 0.7753 | 0.6773 | 0.8772 |

The method developed in this study was subjected to the Wilcoxon signed-rank test separately with each method used for comparison. Cases with $p < 0.05$ as a result of the test indicate that there is a significant difference between the methods compared. The cases where the Z value is negative (−) mean that there is a difference in favor of the developed method. Pearson correlation coefficient (r) expresses the effect value. A Pearson correlation coefficient of 0.1 is considered a small effect, 0.3 is considered a medium effect, and 0.5 is considered a large effect (*Cevahir, 2020*).

When Table 13 is examined, it can be seen that there is a significant difference in favor of the developed method in all comparisons of PSNR and SSIM between the developed method and the compared methods. In IEF comparisons, it can be seen that there is a significant difference in favor of the method developed in the others, except for one. It is

**Table 6 Mean IEF results for the 200 Berkeley dataset (BSDS) images with different SPN ratios.** AWMF was realized with 199 images. (AWMF was unable to process image 292066 in the Berkeley dataset).

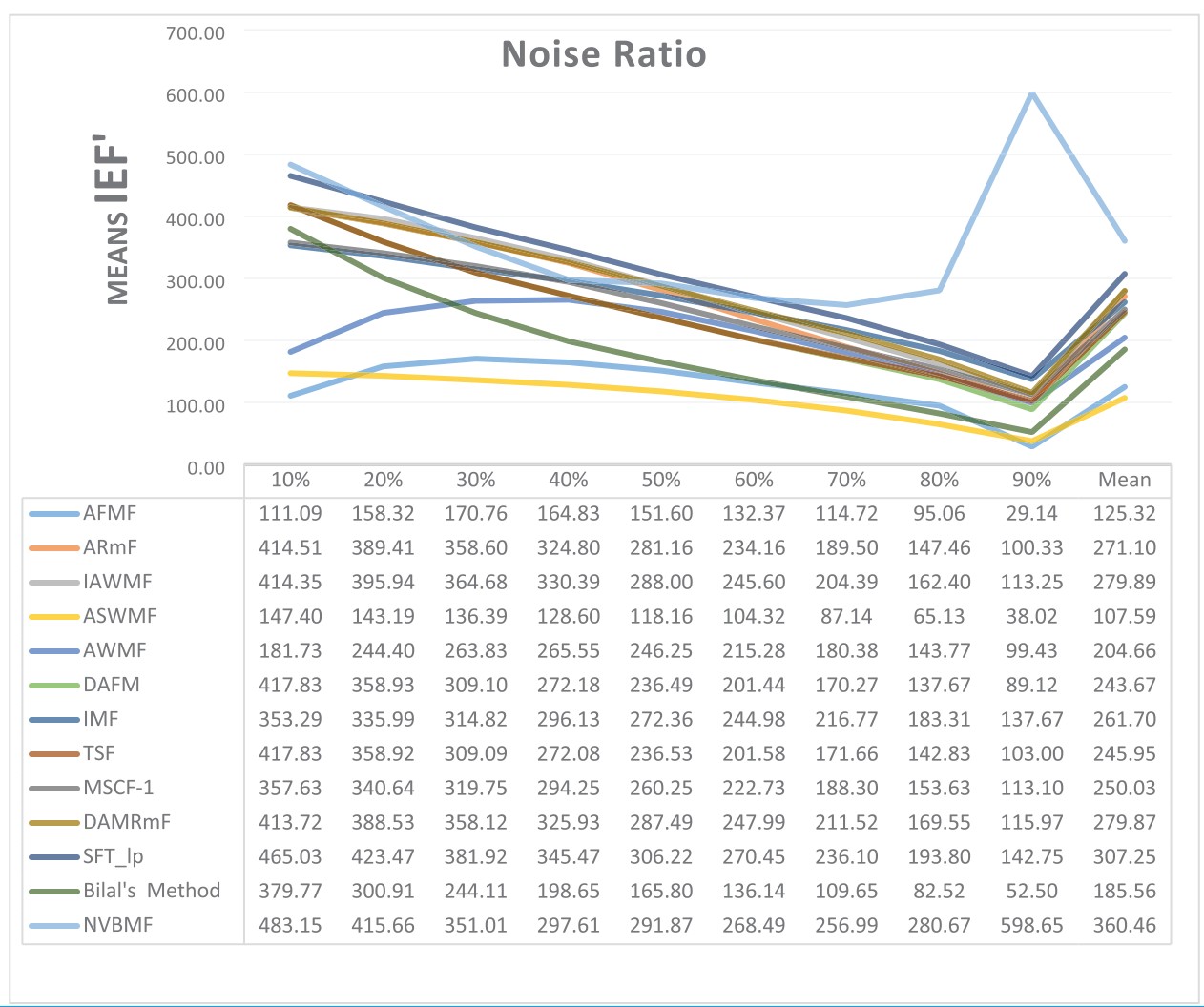

| | 10% | 20% | 30% | 40% | 50% | 60% | 70% | 80% | 90% | Mean |
|---|---|---|---|---|---|---|---|---|---|---|
| AFMF | 111.09 | 158.32 | 170.76 | 164.83 | 151.60 | 132.37 | 114.72 | 95.06 | 29.14 | 125.32 |
| ARmF | 414.51 | 389.41 | 358.60 | 324.80 | 281.16 | 234.16 | 189.50 | 147.46 | 100.33 | 271.10 |
| IAWMF | 414.35 | 395.94 | 364.68 | 330.39 | 288.00 | 245.60 | 204.39 | 162.40 | 113.25 | 279.89 |
| ASWMF | 147.40 | 143.19 | 136.39 | 128.60 | 118.16 | 104.32 | 87.14 | 65.13 | 38.02 | 107.59 |
| AWMF | 181.73 | 244.40 | 263.83 | 265.55 | 246.25 | 215.28 | 180.38 | 143.77 | 99.43 | 204.66 |
| DAFM | 417.83 | 358.93 | 309.10 | 272.18 | 236.49 | 201.44 | 170.27 | 137.67 | 89.12 | 243.67 |
| IMF | 353.29 | 335.99 | 314.82 | 296.13 | 272.36 | 244.98 | 216.77 | 183.31 | 137.67 | 261.70 |
| TSF | 417.83 | 358.92 | 309.09 | 272.08 | 236.53 | 201.58 | 171.66 | 142.83 | 103.00 | 245.95 |
| MSCF-1 | 357.63 | 340.64 | 319.75 | 294.25 | 260.25 | 222.73 | 188.30 | 153.63 | 113.10 | 250.03 |
| DAMRmF | 413.72 | 388.53 | 358.12 | 325.93 | 287.49 | 247.99 | 211.52 | 169.55 | 115.97 | 279.87 |
| SFT_lp | 465.03 | 423.47 | 381.92 | 345.47 | 306.22 | 270.45 | 236.10 | 193.80 | 142.75 | 307.25 |
| Bilal's Method | 379.77 | 300.91 | 244.11 | 198.65 | 165.80 | 136.14 | 109.65 | 82.52 | 52.50 | 185.56 |
| NVBMF | 483.15 | 415.66 | 351.01 | 297.61 | 291.87 | 268.49 | 256.99 | 280.67 | 598.65 | 360.46 |

seen that a significant difference creates a small effect in the IAWMF SSIM comparison, a significant difference creates a medium effect in the IAWMF IEF, DAMRmF comparisons. No significant difference was found in the DAMRmF IEF comparison. And a significant difference creates a large effect in all other comparisons.

When Table 14 is examined, it can be seen that there is a significant difference in favor of the developed method in all comparisons of SSIM, and IEF between the developed method and the compared methods. In PSNR comparisons, it can be seen that there is a significant difference in favor of the method developed in the others, except for one. In comparisons of IAWMF SSIM, IAWMF IEF, SFT_lp SSIM and ARmF IEF, it can be seen that a significant difference creates a medium effect. In comparisons of SFT_lp IEF, it can be seen that a significant difference creates a small effect. No significant difference was

**Table 7 Mean PSNR results for the 40 TESTIMAGES dataset with different SPN ratios.**

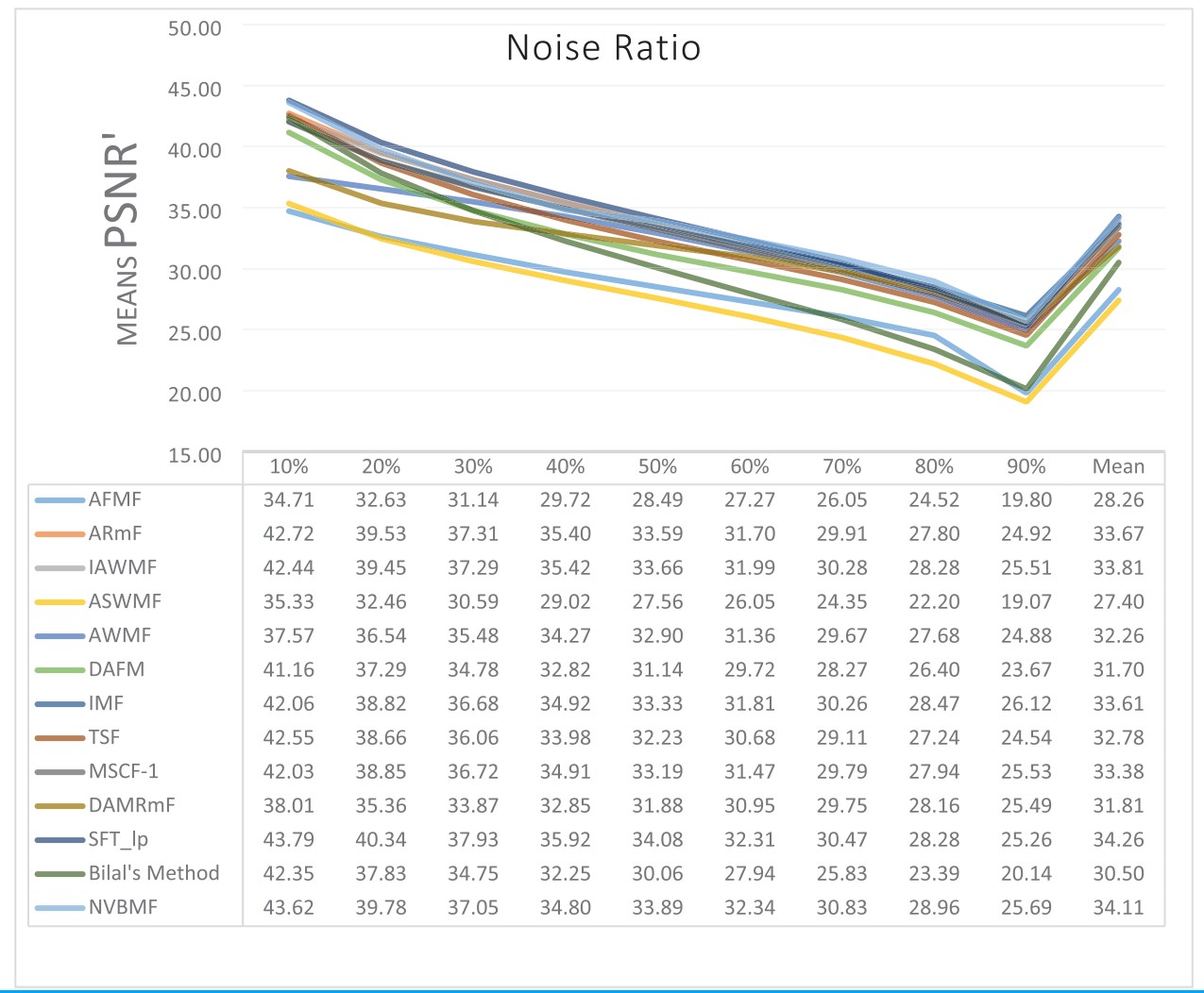

| | 10% | 20% | 30% | 40% | 50% | 60% | 70% | 80% | 90% | Mean |
|---|---|---|---|---|---|---|---|---|---|---|
| AFMF | 34.71 | 32.63 | 31.14 | 29.72 | 28.49 | 27.27 | 26.05 | 24.52 | 19.80 | 28.26 |
| ARmF | 42.72 | 39.53 | 37.31 | 35.40 | 33.59 | 31.70 | 29.91 | 27.80 | 24.92 | 33.67 |
| IAWMF | 42.44 | 39.45 | 37.29 | 35.42 | 33.66 | 31.99 | 30.28 | 28.28 | 25.51 | 33.81 |
| ASWMF | 35.33 | 32.46 | 30.59 | 29.02 | 27.56 | 26.05 | 24.35 | 22.20 | 19.07 | 27.40 |
| AWMF | 37.57 | 36.54 | 35.48 | 34.27 | 32.90 | 31.36 | 29.67 | 27.68 | 24.88 | 32.26 |
| DAFM | 41.16 | 37.29 | 34.78 | 32.82 | 31.14 | 29.72 | 28.27 | 26.40 | 23.67 | 31.70 |
| IMF | 42.06 | 38.82 | 36.68 | 34.92 | 33.33 | 31.81 | 30.26 | 28.47 | 26.12 | 33.61 |
| TSF | 42.55 | 38.66 | 36.06 | 33.98 | 32.23 | 30.68 | 29.11 | 27.24 | 24.54 | 32.78 |
| MSCF-1 | 42.03 | 38.85 | 36.72 | 34.91 | 33.19 | 31.47 | 29.79 | 27.94 | 25.53 | 33.38 |
| DAMRmF | 38.01 | 35.36 | 33.87 | 32.85 | 31.88 | 30.95 | 29.75 | 28.16 | 25.49 | 31.81 |
| SFT_lp | 43.79 | 40.34 | 37.93 | 35.92 | 34.08 | 32.31 | 30.47 | 28.28 | 25.26 | 34.26 |
| Bilal's Method | 42.35 | 37.83 | 34.75 | 32.25 | 30.06 | 27.94 | 25.83 | 23.39 | 20.14 | 30.50 |
| NVBMF | 43.62 | 39.78 | 37.05 | 34.80 | 33.89 | 32.34 | 30.83 | 28.96 | 25.69 | 34.11 |

found in the SFT_lp PSNR comparison. And a significant difference creates a large effect in all other comparisons.

When Table 15 is examined, it can be seen that there is a difference in favor of the method developed according to Z values in all comparisons, except SFT_lp IEF. However, some of them were found to have significant differences. From PSNR comparisons, it can be seen that DAMF has a small effect, AFMF and SFT_lp have a medium effect, ASWMF, MSCF-1, DAMRmF, Bilal's Method have a large effect. From the SSIM comparisons, it can be seen that the IMF has a small effect, AWMF-DAMF-TSF have a medium effect, and AFMF-ASWMF-MSCF-1-DAMRmF-SFT_lp-Bilal's Method have a large effect. From the IEF comparisons, it can be seen that AWMF-DAMF-TSF has a small effect, DAMRmF has a medium effect and AFMF-ASWMFMSCF-1-Bilal's Mehod have a large effect. In the only

**Table 8 Mean SSIM results for the 40 TESTIMAGES dataset with different SPN ratios.**

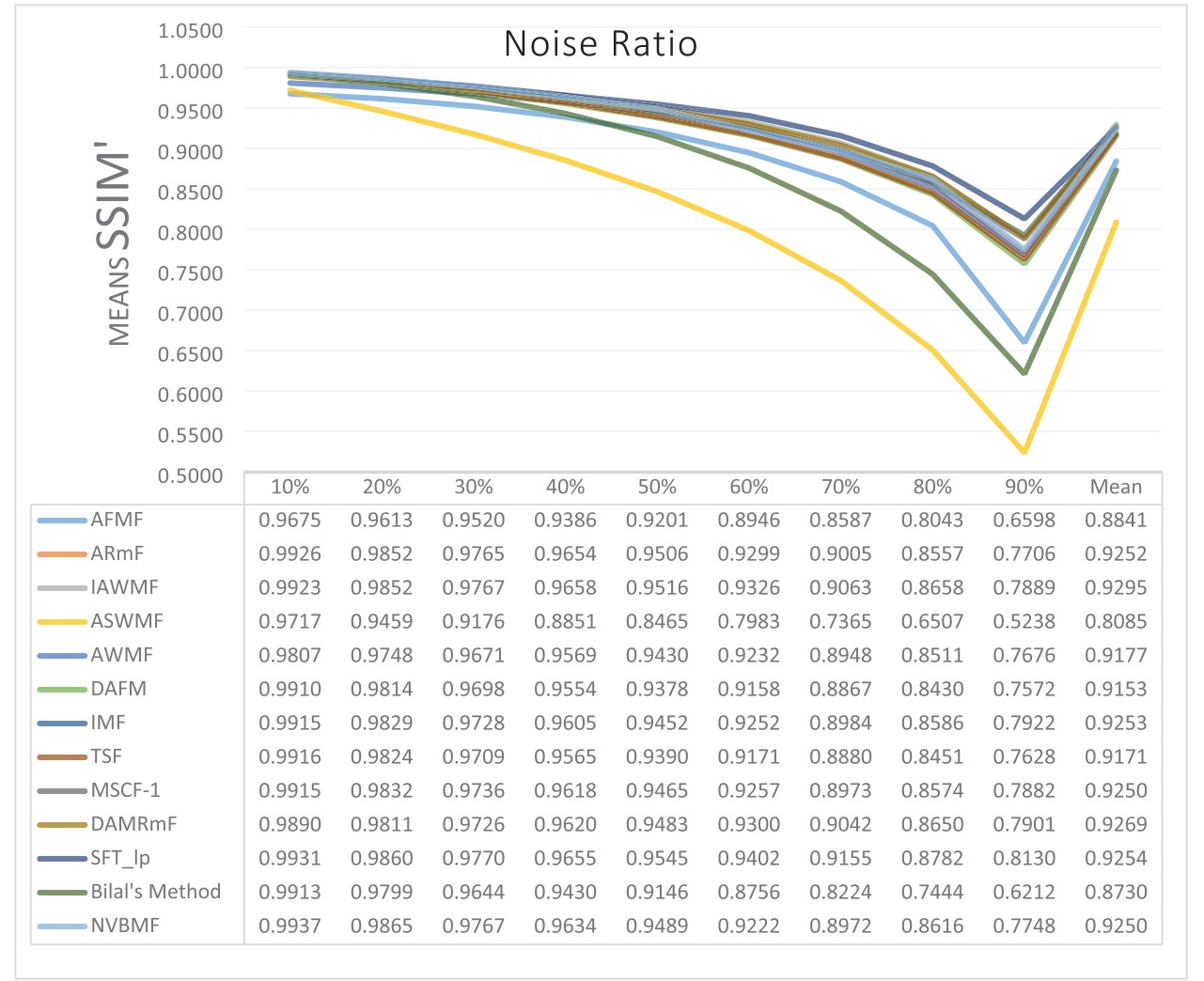

| | 10% | 20% | 30% | 40% | 50% | 60% | 70% | 80% | 90% | Mean |
|---|---|---|---|---|---|---|---|---|---|---|
| AFMF | 0.9675 | 0.9613 | 0.9520 | 0.9386 | 0.9201 | 0.8946 | 0.8587 | 0.8043 | 0.6598 | 0.8841 |
| ARmF | 0.9926 | 0.9852 | 0.9765 | 0.9654 | 0.9506 | 0.9299 | 0.9005 | 0.8557 | 0.7706 | 0.9252 |
| IAWMF | 0.9923 | 0.9852 | 0.9767 | 0.9658 | 0.9516 | 0.9326 | 0.9063 | 0.8658 | 0.7889 | 0.9295 |
| ASWMF | 0.9717 | 0.9459 | 0.9176 | 0.8851 | 0.8465 | 0.7983 | 0.7365 | 0.6507 | 0.5238 | 0.8085 |
| AWMF | 0.9807 | 0.9748 | 0.9671 | 0.9569 | 0.9430 | 0.9232 | 0.8948 | 0.8511 | 0.7676 | 0.9177 |
| DAFM | 0.9910 | 0.9814 | 0.9698 | 0.9554 | 0.9378 | 0.9158 | 0.8867 | 0.8430 | 0.7572 | 0.9153 |
| IMF | 0.9915 | 0.9829 | 0.9728 | 0.9605 | 0.9452 | 0.9252 | 0.8984 | 0.8586 | 0.7922 | 0.9253 |
| TSF | 0.9916 | 0.9824 | 0.9709 | 0.9565 | 0.9390 | 0.9171 | 0.8880 | 0.8451 | 0.7628 | 0.9171 |
| MSCF-1 | 0.9915 | 0.9832 | 0.9736 | 0.9618 | 0.9465 | 0.9257 | 0.8973 | 0.8574 | 0.7882 | 0.9250 |
| DAMRmF | 0.9890 | 0.9811 | 0.9726 | 0.9620 | 0.9483 | 0.9300 | 0.9042 | 0.8650 | 0.7901 | 0.9269 |
| SFT_lp | 0.9931 | 0.9860 | 0.9770 | 0.9655 | 0.9545 | 0.9402 | 0.9155 | 0.8782 | 0.8130 | 0.9254 |
| Bilal's Method | 0.9913 | 0.9799 | 0.9644 | 0.9430 | 0.9146 | 0.8756 | 0.8224 | 0.7444 | 0.6212 | 0.8730 |
| NVBMF | 0.9937 | 0.9865 | 0.9767 | 0.9634 | 0.9489 | 0.9222 | 0.8972 | 0.8616 | 0.7748 | 0.9250 |

SFT_lp IEF comparison, a small significant difference is in favor of SFT_lp. The effect rates in other comparisons were not found to be significant.

It has been determined that there are differences in favor of the developed method in all of the comparisons except for three. According to the results obtained in the Berkeley dataset and the significance test results: Significant differences were found in favor of the developed method in all of the significance tests for PSNR and SSIM. Significant differences were found in 11 of 12 tests for IEF results. Only one found a difference in favor of SFT_lp, but this difference was not significant (does not meet the $p < 0.05$) condition. According to the results obtained in the TESTIMAGES dataset and the significance test results: There were significant differences in favor of the developed method in all of the significance tests for SSIM. Differences were found in favor of the developed method in all

**Table 9 Mean IEF results for the 40 TESTIMAGES dataset with different SPN ratios.**

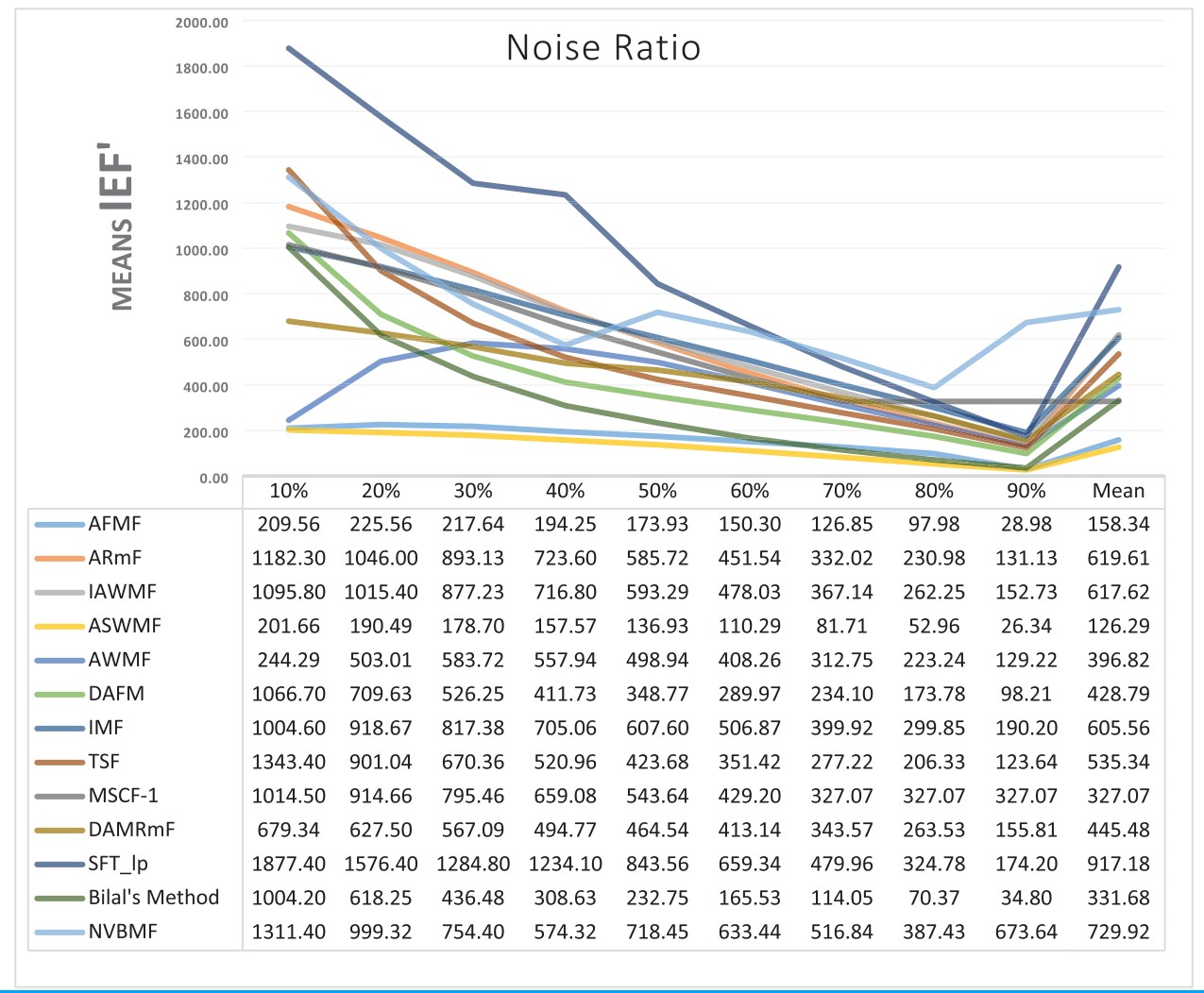

| | 10% | 20% | 30% | 40% | 50% | 60% | 70% | 80% | 90% | Mean |
|---|---|---|---|---|---|---|---|---|---|---|
| AFMF | 209.56 | 225.56 | 217.64 | 194.25 | 173.93 | 150.30 | 126.85 | 97.98 | 28.98 | 158.34 |
| ARmF | 1182.30 | 1046.00 | 893.13 | 723.60 | 585.72 | 451.54 | 332.02 | 230.98 | 131.13 | 619.61 |
| IAWMF | 1095.80 | 1015.40 | 877.23 | 716.80 | 593.29 | 478.03 | 367.14 | 262.25 | 152.73 | 617.62 |
| ASWMF | 201.66 | 190.49 | 178.70 | 157.57 | 136.93 | 110.29 | 81.71 | 52.96 | 26.34 | 126.29 |
| AWMF | 244.29 | 503.01 | 583.72 | 557.94 | 498.94 | 408.26 | 312.75 | 223.24 | 129.22 | 396.82 |
| DAFM | 1066.70 | 709.63 | 526.25 | 411.73 | 348.77 | 289.97 | 234.10 | 173.78 | 98.21 | 428.79 |
| IMF | 1004.60 | 918.67 | 817.38 | 705.06 | 607.60 | 506.87 | 399.92 | 299.85 | 190.20 | 605.56 |
| TSF | 1343.40 | 901.04 | 670.36 | 520.96 | 423.68 | 351.42 | 277.22 | 206.33 | 123.64 | 535.34 |
| MSCF-1 | 1014.50 | 914.66 | 795.46 | 659.08 | 543.64 | 429.20 | 327.07 | 327.07 | 327.07 | 327.07 |
| DAMRmF | 679.34 | 627.50 | 567.09 | 494.77 | 464.54 | 413.14 | 343.57 | 263.53 | 155.81 | 445.48 |
| SFT_lp | 1877.40 | 1576.40 | 1284.80 | 1234.10 | 843.56 | 659.34 | 479.96 | 324.78 | 174.20 | 917.18 |
| Bilal's Method | 1004.20 | 618.25 | 436.48 | 308.63 | 232.75 | 165.53 | 114.05 | 70.37 | 34.80 | 331.68 |
| NVBMF | 1311.40 | 999.32 | 754.40 | 574.32 | 718.45 | 633.44 | 516.84 | 387.43 | 673.64 | 729.92 |

of the tests performed for PSNR. Except for one of these differences (does not meet the $p < 0.05$) condition, 11 are significant. Significant differences were found in favor of the method developed in 11 of the significance tests for IEF and against the method developed in one (SFT_lp). According to the significance test results for MATLAB library images: There were differences in favor of the developed method in all tests for PSNR. Except for five of these differences (does not meet the $p < 0.05$) condition, seven of them are significant. Differences were found in favor of the developed method in all of the tests performed for SSIM. Except for two of these differences (does not meet the $p < 0.05$) condition, 10 are significant. There were differences in favor of the developed method in eleven of the tests performed for IEF. All of these differences are significant. A significant difference was found against the developed method in one of them (SFT_lp).

**Table 10 Mean PSNR results for the 20 MATLAB library images with different SPN ratios.**

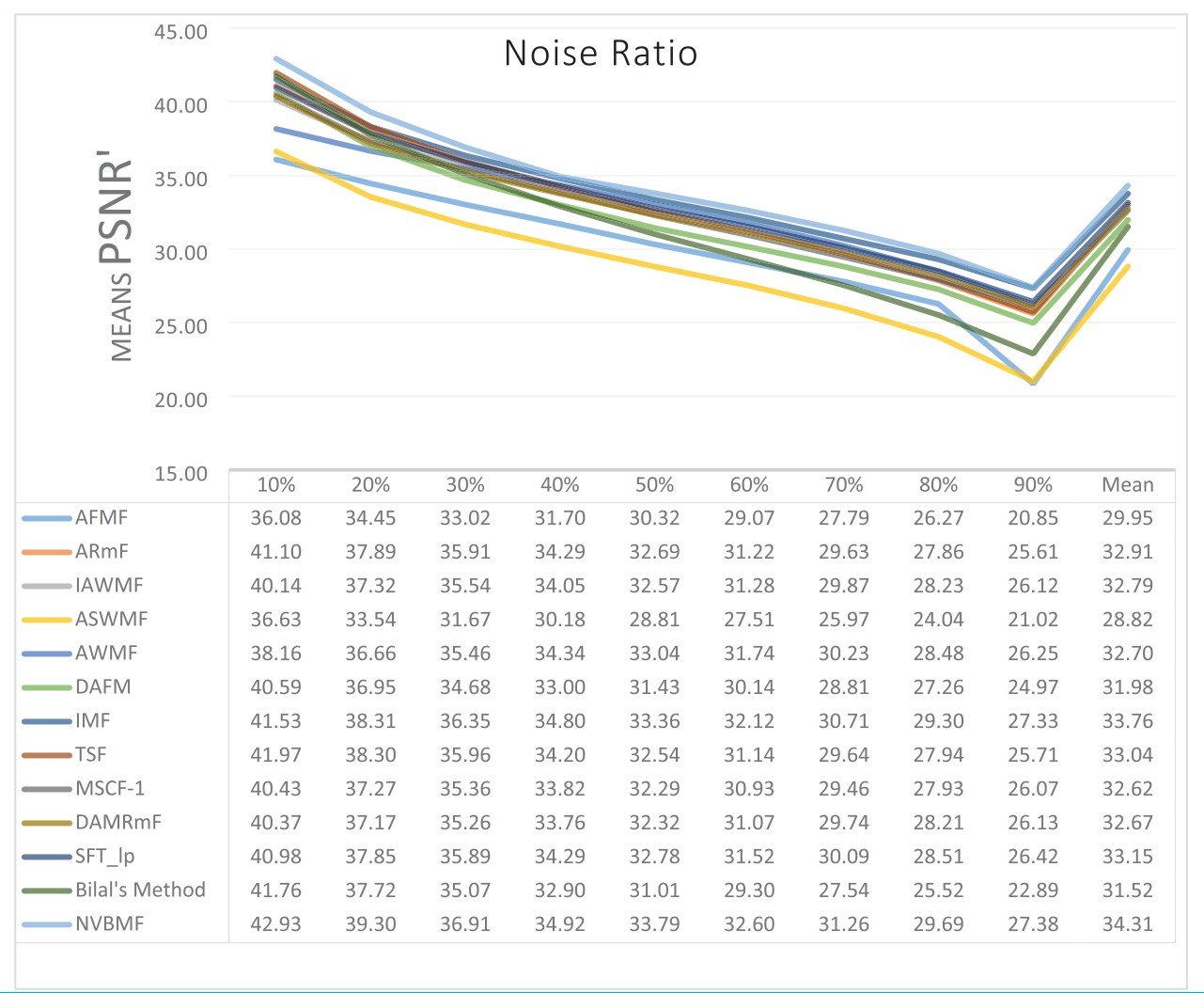

| | 10% | 20% | 30% | 40% | 50% | 60% | 70% | 80% | 90% | Mean |
|---|---|---|---|---|---|---|---|---|---|---|
| AFMF | 36.08 | 34.45 | 33.02 | 31.70 | 30.32 | 29.07 | 27.79 | 26.27 | 20.85 | 29.95 |
| ARmF | 41.10 | 37.89 | 35.91 | 34.29 | 32.69 | 31.22 | 29.63 | 27.86 | 25.61 | 32.91 |
| IAWMF | 40.14 | 37.32 | 35.54 | 34.05 | 32.57 | 31.28 | 29.87 | 28.23 | 26.12 | 32.79 |
| ASWMF | 36.63 | 33.54 | 31.67 | 30.18 | 28.81 | 27.51 | 25.97 | 24.04 | 21.02 | 28.82 |
| AWMF | 38.16 | 36.66 | 35.46 | 34.34 | 33.04 | 31.74 | 30.23 | 28.48 | 26.25 | 32.70 |
| DAFM | 40.59 | 36.95 | 34.68 | 33.00 | 31.43 | 30.14 | 28.81 | 27.26 | 24.97 | 31.98 |
| IMF | 41.53 | 38.31 | 36.35 | 34.80 | 33.36 | 32.12 | 30.71 | 29.30 | 27.33 | 33.76 |
| TSF | 41.97 | 38.30 | 35.96 | 34.20 | 32.54 | 31.14 | 29.64 | 27.94 | 25.71 | 33.04 |
| MSCF-1 | 40.43 | 37.27 | 35.36 | 33.82 | 32.29 | 30.93 | 29.46 | 27.93 | 26.07 | 32.62 |
| DAMRmF | 40.37 | 37.17 | 35.26 | 33.76 | 32.32 | 31.07 | 29.74 | 28.21 | 26.13 | 32.67 |
| SFT_lp | 40.98 | 37.85 | 35.89 | 34.29 | 32.78 | 31.52 | 30.09 | 28.51 | 26.42 | 33.15 |
| Bilal's Method | 41.76 | 37.72 | 35.07 | 32.90 | 31.01 | 29.30 | 27.54 | 25.52 | 22.89 | 31.52 |
| NVBMF | 42.93 | 39.30 | 36.91 | 34.92 | 33.79 | 32.60 | 31.26 | 29.69 | 27.38 | 34.31 |

A total of 108 comparisons were made for significance tests. A difference was found in favor of the developed method in 105 of them. A total of 94 of the differences found in favor of the developed method are significant, 11 of them are not. Three differences were found against the developed method. Two of them are significant, one of them is not.

The differences against the developed method are the significance tests with the SFT_lp method. For this reason, there are nine tests in total when SFT_lp and the developed method significance tests are compared one to one. A big difference in favor of the Berkeley PSNR developed method, a big difference in favor of the Berkeley SSIM developed method, and a nonsignificant difference in favor of the Berkeley IEF SFT_lp were determined. Insignificant difference in favor of TESTIMAGES PSNR developed

**Table 11 Mean SSIM results for the 20 MATLAB library images with different SPN ratios.**

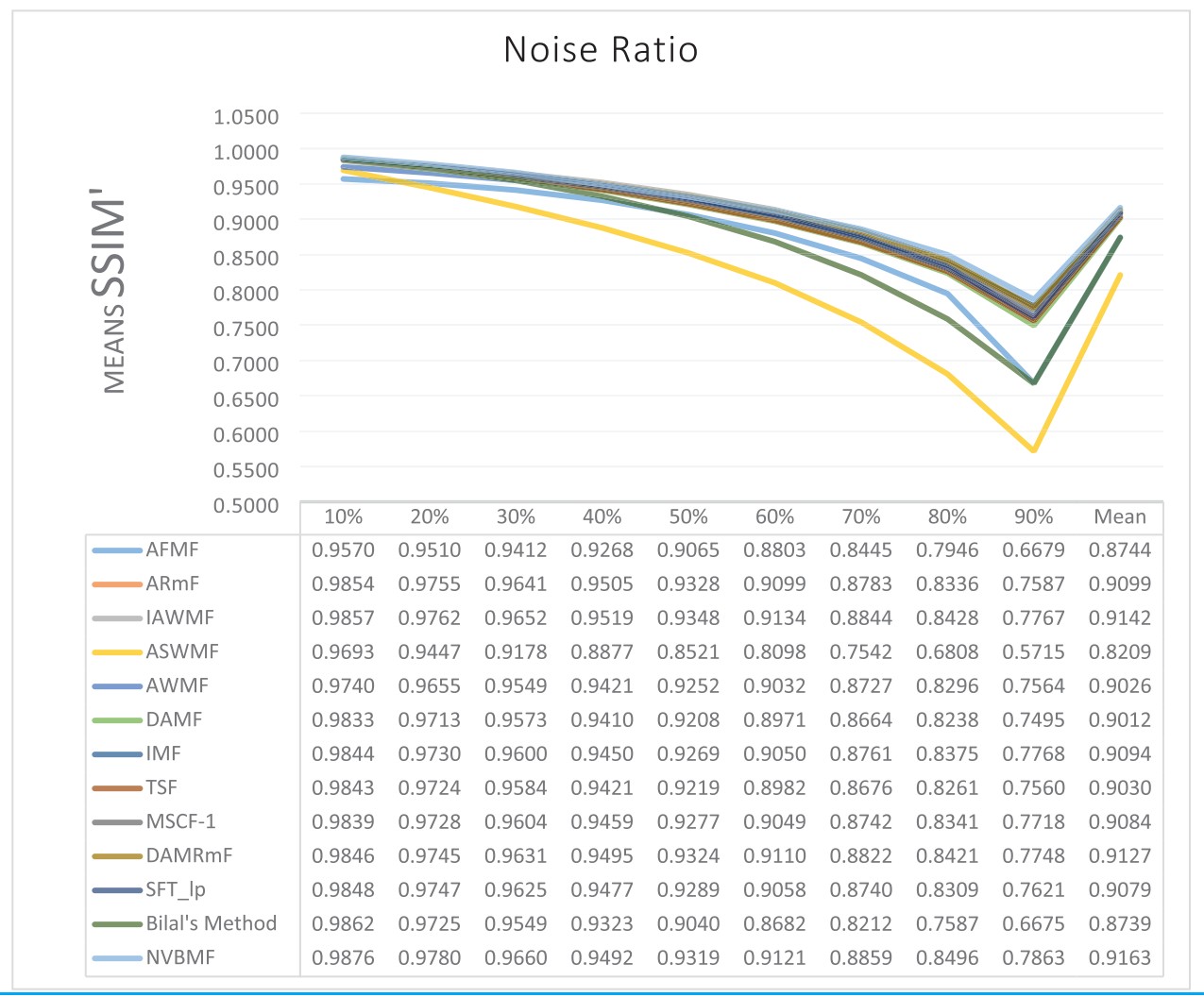

| | 10% | 20% | 30% | 40% | 50% | 60% | 70% | 80% | 90% | Mean |
|---|---|---|---|---|---|---|---|---|---|---|
| AFMF | 0.9570 | 0.9510 | 0.9412 | 0.9268 | 0.9065 | 0.8803 | 0.8445 | 0.7946 | 0.6679 | 0.8744 |
| ARmF | 0.9854 | 0.9755 | 0.9641 | 0.9505 | 0.9328 | 0.9099 | 0.8783 | 0.8336 | 0.7587 | 0.9099 |
| IAWMF | 0.9857 | 0.9762 | 0.9652 | 0.9519 | 0.9348 | 0.9134 | 0.8844 | 0.8428 | 0.7767 | 0.9142 |
| ASWMF | 0.9693 | 0.9447 | 0.9178 | 0.8877 | 0.8521 | 0.8098 | 0.7542 | 0.6808 | 0.5715 | 0.8209 |
| AWMF | 0.9740 | 0.9655 | 0.9549 | 0.9421 | 0.9252 | 0.9032 | 0.8727 | 0.8296 | 0.7564 | 0.9026 |
| DAMF | 0.9833 | 0.9713 | 0.9573 | 0.9410 | 0.9208 | 0.8971 | 0.8664 | 0.8238 | 0.7495 | 0.9012 |
| IMF | 0.9844 | 0.9730 | 0.9600 | 0.9450 | 0.9269 | 0.9050 | 0.8761 | 0.8375 | 0.7768 | 0.9094 |
| TSF | 0.9843 | 0.9724 | 0.9584 | 0.9421 | 0.9219 | 0.8982 | 0.8676 | 0.8261 | 0.7560 | 0.9030 |
| MSCF-1 | 0.9839 | 0.9728 | 0.9604 | 0.9459 | 0.9277 | 0.9049 | 0.8742 | 0.8341 | 0.7718 | 0.9084 |
| DAMRmF | 0.9846 | 0.9745 | 0.9631 | 0.9495 | 0.9324 | 0.9110 | 0.8822 | 0.8421 | 0.7748 | 0.9127 |
| SFT_lp | 0.9848 | 0.9747 | 0.9625 | 0.9477 | 0.9289 | 0.9058 | 0.8740 | 0.8309 | 0.7621 | 0.9079 |
| Bilal's Method | 0.9862 | 0.9725 | 0.9549 | 0.9323 | 0.9040 | 0.8682 | 0.8212 | 0.7587 | 0.6675 | 0.8739 |
| NVBMF | 0.9876 | 0.9780 | 0.9660 | 0.9492 | 0.9319 | 0.9121 | 0.8859 | 0.8496 | 0.7863 | 0.9163 |

method, moderate difference in favor of TESTIMAGES SSIM developed method, small difference in favor of TESTIMAGES IEF SFT_lp. MATLAB library images have a moderate difference in favor of the PSNR developed method, MATLAB library images have a large difference in favor of the SSIM developed method, MATLAB library images have a small difference in favor of IEF SFT_lp. In total, five of nine comparisons contain significant differences in favor of the developed method. Two of them contain a significant difference in favor of SFT_lp. The other two comparisons do not contain significant difference.

In this case, it can be said that the results obtained with the developed method are better than the compared methods.

**Table 12 Mean IEF results for the 20 MATLAB library images with different SPN ratios.**

| | 10% | 20% | 30% | 40% | 50% | 60% | 70% | 80% | 90% | Mean |
|---|---|---|---|---|---|---|---|---|---|---|
| AFMF | 245.08 | 326.91 | 340.04 | 324.59 | 286.69 | 247.95 | 211.29 | 163.37 | 39.53 | 242.83 |
| ARmF | 1019.79 | 915.14 | 816.77 | 729.63 | 605.86 | 496.63 | 389.73 | 283.30 | 176.58 | 603.71 |
| IAWMF | 989.98 | 916.18 | 822.88 | 737.66 | 619.24 | 519.80 | 422.13 | 313.55 | 200.00 | 615.71 |
| ASWMF | 249.00 | 239.55 | 228.31 | 215.42 | 195.43 | 171.79 | 142.97 | 104.96 | 67.10 | 179.90 |
| AWMF | 347.22 | 497.82 | 551.17 | 564.07 | 513.51 | 446.26 | 364.68 | 272.83 | 173.39 | 414.55 |
| DAFM | 1007.80 | 774.76 | 628.29 | 544.59 | 456.00 | 393.12 | 332.77 | 255.99 | 154.11 | 505.27 |
| IMF | 844.87 | 766.81 | 699.21 | 644.75 | 565.96 | 495.42 | 420.44 | 330.70 | 226.84 | 555.00 |
| TSF | 1007.80 | 774.76 | 628.28 | 544.54 | 456.22 | 393.32 | 334.73 | 261.38 | 172.40 | 508.16 |
| MSCF-1 | 874.50 | 796.78 | 724.21 | 659.45 | 560.52 | 470.77 | 381.14 | 295.42 | 203.18 | 551.78 |
| DAMRmF | 1019.50 | 912.95 | 813.02 | 731.98 | 618.65 | 522.40 | 426.80 | 321.23 | 205.22 | 619.08 |
| SFT_lp | 1310.70 | 1099.30 | 940.80 | 824.60 | 685.13 | 573.78 | 463.91 | 343.18 | 217.11 | 717.62 |
| Bilal's Method | 853.92 | 603.78 | 470.81 | 364.67 | 289.31 | 220.42 | 171.84 | 113.72 | 63.31 | 350.20 |
| NVBMF | 1175.10 | 945.66 | 764.92 | 634.81 | 639.89 | 571.34 | 486.94 | 377.19 | 237.72 | 648.18 |

**Table 13 Wilcoxon signed-rank test and Pearson correlation coefficient of the results obtained with 200 Berkeley dataset (BSDS).** Decimal full values were used in the calculations.

| $n = 1{,}800$ | NVBMF | | | | | | | | |
|---|---|---|---|---|---|---|---|---|---|
| | **PSNR** | | | **SSIM** | | | **IEF** | | |
| | $p$ | **Z value** | $r(Z/\sqrt{n})$ | $p$ | **Z value** | $r(Z/\sqrt{n})$ | $p$ | **Z value** | $r(Z/\sqrt{n})$ |
| AFMF | $2.49 \times 10^{-279}$ | $-35.7129$ | 0.84 | $4.95 \times 10^{-284}$ | $-36.0145$ | 0.85 | $1.28 \times 10^{-295}$ | $-36.7474$ | 0.87 |
| ARmF | $1.56 \times 10^{-208}$ | $-30.8168$ | 0.73 | $2.15 \times 10^{-123}$ | $-23.6246$ | 0.56 | $1.27 \times 10^{-134}$ | $-24.6929$ | 0.58 |
| IAWMF | $9.38 \times 10^{-132}$ | $-24.4246$ | 0.58 | $1.29 \times 10^{-5}$ | $-4.3616$ | 0.10 | $1.34 \times 10^{-76}$ | $-18.5231$ | 0.44 |
| ASWMF | $5.40 \times 10^{-279}$ | $-35.6913$ | 0.84 | $3.81 \times 10^{-283}$ | $-35.9579$ | 0.85 | $1.29 \times 10^{-295}$ | $-36.7473$ | 0.87 |
| AWMF | $1.91 \times 10^{-290}$ | $-36.4213$ | 0.86 | $1.96 \times 10^{-290}$ | $-36.4220$ | 0.86 | $4.29 \times 10^{-294}$ | $-36.6517$ | 0.86 |
| DAMF | $1.1 \times 10^{-278}$ | $-35.6711$ | 0.84 | $3.79 \times 10^{-283}$ | $-35.9580$ | 0.85 | $6.88 \times 10^{-263}$ | $-34.6375$ | 0.82 |
| IMF | $6.19 \times 10^{-223}$ | $-31.8737$ | 0.75 | $3.85 \times 10^{-275}$ | $-35.4420$ | 0.84 | $4.23 \times 10^{-168}$ | $-27.6350$ | 0.65 |
| TSF | $1.89 \times 10^{-278}$ | $-35.6562$ | 0.84 | $5.63 \times 10^{-279}$ | $-35.6901$ | 0.84 | $2.62 \times 10^{-262}$ | $-34.5990$ | 0.82 |
| MSCF-1 | $1.64 \times 10^{-278}$ | -35,6602 | 0.84 | $5.58 \times 10^{-281}$ | $-35.8191$ | 0.84 | $2.20 \times 10^{-276}$ | $-35.5277$ | 0.84 |
| DAMRmF | $1.13 \times 10^{-181}$ | $-28.7424$ | 0.68 | $5.94 \times 10^{-65}$ | $-17.0190$ | 0.40 | $1.02 \times 10^{-117}$ | $-23.0659$ | 0.54 |
| SFT_lp | $2.52 \times 10^{-176}$ | $-28.3112$ | 0.67 | $4.63 \times 10^{-208}$ | $-30.7815$ | 0,73 | $3.43 \times 10^{-1}$ | 0.9489 | — |
| Bilal's method | $2.95 \times 10^{-284}$ | $-36.0289$ | 0.85 | $4.95 \times 10^{-284}$ | $-36.0145$ | 0.85 | $1.63 \times 10^{-293}$ | $-36.6154$ | 0.86 |

**Table 14 Wilcoxon signed-rank test and Pearson correlation coefficient of the results obtained with 40 TESTIMAGES dataset.** Decimal full values were used in the calculations.

| $n = 360$ | NVBMF | | | | | | | | |
|---|---|---|---|---|---|---|---|---|---|
| | **PSNR** | | | **SSIM** | | | **IEF** | | |
| | $p$ | **Z value** | $r(Z/\sqrt{n})$ | $p$ | **Z value** | $r(Z/\sqrt{n})$ | $p$ | **Z value** | $r(Z/\sqrt{n})$ |
| AFMF | $8.08 \times 10^{-55}$ | $-15.5933$ | 0.82 | $1.91 \times 10^{-53}$ | $-15.3899$ | 0.81 | $9.40 \times 10^{-61}$ | $-16.4431$ | 0.87 |
| ARmF | $3.03 \times 10^{-29}$ | $-11.2262$ | 0.59 | $1.80 \times 10^{-28}$ | $-11.0678$ | 0.58 | $3.58 \times 10^{-15}$ | $-7.8688$ | 0.41 |
| IAWMF | $9.65 \times 10^{-27}$ | $-10.7049$ | 0.57 | $1.05 \times 10^{-19}$ | $-9.0834$ | 0.48 | $5.76 \times 10^{-15}$ | $-7.8091$ | 0.41 |
| ASWMF | $6.22 \times 10^{-56}$ | $-15.7563$ | 0.83 | $1.63 \times 10^{-54}$ | $-15.5483$ | 0.82 | $9.40 \times 10^{-61}$ | $-16.4431$ | 0.87 |
| AWMF | $2.87 \times 10^{-53}$ | $-15.3636$ | 0.81 | $3.16 \times 10^{-53}$ | $-15.3575$ | 0.81 | $8.14 \times 10^{-58}$ | $-16.0281$ | 0.84 |
| DAMF | $3.23 \times 10^{-53}$ | $-15.3560$ | 0.81 | $3.16 \times 10^{-53}$ | $-15.3575$ | 0.81 | $1.06 \times 10^{-53}$ | $-15.4278$ | 0.81 |
| IMF | $8.41 \times 10^{-43}$ | $-13.7137$ | 0.72 | $3.15 \times 10^{-48}$ | $-14.5923$ | 0.77 | $5.39 \times 10^{-31}$ | $-11.5770$ | 0.61 |
| TSF | $4.02 \times 10^{-53}$ | $-15.3418$ | 0.81 | $3.16 \times 10^{-53}$ | $-15.3575$ | 0.81 | $1.04 \times 10^{-50}$ | $-14.9769$ | 0.79 |
| MSCF-1 | $6.22 \times 10^{-48}$ | $-14.5457$ | 0.77 | $2.57 \times 10^{-51}$ | $-15.0695$ | 0.79 | $1.63 \times 10^{-53}$ | $-12.4378$ | 0.66 |
| DAMRmF | $3.78 \times 10^{-50}$ | $-14.8909$ | 0.78 | $7.41 \times 10^{-47}$ | $-14.3752$ | 0.76 | $1.64 \times 10^{-40}$ | $-13.3255$ | 0.70 |
| SFT_lp | $3.24 \times 10^{-1}$ | $-0.9854$ | — | $3.32 \times 10^{-15}$ | $-7.8784$ | 0.42 | $4.71 \times 10^{-11}$ | 6.5798 | 0.35 |
| Bilal's method | $1.02 \times 10^{-53}$ | $-15.4304$ | 0.81 | $1.10 \times 10^{-53}$ | $-15.4258$ | 0.81 | $9.96 \times 10^{-61}$ | $-16.4395$ | 0.91 |

**Table 15 Wilcoxon signed-rank test and Pearson correlation coefficient of the results obtained with 20 MATLAB library images.** Decimal full values were used in the calculations.

| $n = 180$ | NVBMF | | | | | | | | |
| --- | --- | --- | --- | --- | --- | --- | --- | --- | --- |
| | PSNR | | | SSIM | | | IEF | | |
| | $p$ | Z value | $r(Z/\sqrt{n})$ | $p$ | Z value | $r(Z/\sqrt{n})$ | $p$ | Z value | $r(Z/\sqrt{n})$ |
| AFMF | $1.56 \times 10^{-09}$ | −6.0382 | 0.45 | $1.67 \times 10^{-14}$ | −7.6738 | 0.57 | $6.57 \times 10^{-12}$ | −6.8667 | 0.51 |
| ARmF | $1.13 \times 10^{-1}$ | −1.5813 | — | $5.17 \times 10^{-2}$ | −1.9456 | — | $5.15 \times 10^{-1}$ | −0.6514 | — |
| IAWMF | $1.67 \times 10^{-1}$ | −1.3823 | — | $4.30 \times 10^{-1}$ | −0.7885 | — | $7.48 \times 10^{-1}$ | −0.3214 | — |
| ASWMF | $5.86 \times 10^{-18}$ | −8.6352 | 0.64 | $1.4410^{-26}$ | −10.6679 | 0.80 | $1.88 \times 10^{-17}$ | −8.5009 | 0.63 |
| AWMF | $8.55 \times 10^{-2}$ | −1.7199 | — | $7.1810^{-6}$ | −4.4883 | 0.33 | $2.10 \times 10^{-3}$ | −3.0712 | 0.23 |
| DAMF | $0.41 \times 10^{-2}$ | −2.8727 | 0.21 | $1.24 \times 10^{-5}$ | −4.3697 | 0.33 | $2.65 \times 10^{-2}$ | −2.2184 | 0.17 |
| IMF | $9.47 \times 10^{-1}$ | −0.0671 | — | $2.00 \times 10^{-3}$ | −3.0898 | 0.23 | $3.14 \times 10^{-1}$ | −1.0071 | — |
| TSF | $2.79 \times 10^{-1}$ | −1.0828 | — | $4.09 \times 10^{-5}$ | −4.1026 | 0.31 | $3.22 \times 10^{-2}$ | −2.1413 | 0.16 |
| MSCF-1 | $3.50 \times 10^{-29}$ | −11.2136 | 0.84 | $1.24 \times 10^{-29}$ | −11.3051 | 0.84 | $3.79 \times 10^{-23}$ | -9,9094 | 0.74 |
| DAMRmF | $5.50 \times 10^{-1}$ | −9.1537 | 0.68 | $4.05 \times 10^{-13}$ | −7.2539 | 0.54 | $1.18 \times 10^{-8}$ | −5.7025 | 0.43 |
| SFT_lp | $4.82 \times 10^{-5}$ | −4.0640 | 0.30 | $2.19 \times 10^{-13}$ | −7.3367 | 0.55 | $8.29 \times 10^{-2}$ | 1.7342 | 0.13 |
| Bilal's Method | $3.95 \times 10^{-31}$ | −11.6036 | 0.86 | $8.78 \times 10^{-31}$ | −11.5350 | 0.86 | $9.08 \times 10^{-31}$ | −11.5322 | 0.86 |

## CONCLUSIONS

An SPN filter based on the nearest pixel values is proposed in the study. NVBMF consists of two phases and uses $11 \times 11$ windows in the first phase. For the noisy center pixel, the nearest noiseless pixel value (averaged if more than one pixel with the same distance exists) is assigned. In the second phase, if the level of noise is more than 45%, also $3 \times 3$ avarage filter is used. In this phase, pixels that are not 0 are averaged as the center pixel value. The NVBMF (proposed) method implemented in two stages has yielded very good results in all levels of noise. In addition, NVBMF was compared with SPN filters developed recently and better results were obtained in many comparisons according to these methods.

In addition, statistical tests show that the differences between the developed method and the compared methods are significant. A significant difference was found in 94 of 108 comparisons.

The use of image processing techniques and obtaining good results are positively affected by successful pre-processing and filter operations on images. In this respect, it is important to develop image filters that produce successful results. It is thought that the SPN removal filter developed in the study will contribute to the literature in this respect.

In the future, the focus is on developing filters based on machine learning to denoise SPN.

### Funding

The author received no funding for this work.

## Competing Interests
The author declares that they have no competing interests.

## Author Contributions
- Bülent Turan conceived and designed the experiments, performed the experiments, analyzed the data, performed the computation work, prepared figures and/or tables, authored or reviewed drafts of the article, and approved the final draft.

## Patent Disclosures
The following patent dependencies were disclosed by the authors:

Data is available from:

1. UC-Berkeley dataset (BSDS)–200 images (*Arbelaez, Fowlkes & Martin, 2007*): https://www2.eecs.berkeley.edu/Research/Projects/CS/vision/bsds/.

2. TESTIMAGES dataset–40 images (*Asuni & Giachetti, 2015*): https://doi.org/10.1080/2165347X.2015.1024298.

3. MATLAB library images–20 images (R2020b; autumn, baby, board, micromarket, car1, coloredChips, fabric, foggyroad, foggysf1, foosball, football, greens, gantrycrane, trailer, hallway, hands1, pears, kobi, lighthouse, onion): MATLAB\R2020b\toolbox\images\imdata.

MATLAB Library images can be accessed from a computer with the Matlab R2020b program installed using the given directory: (MATLAB\R2020b\toolbox\images\imdata).

## Data Availability
The code is available in the Supplemental Files.

## Supplemental Information
Supplemental information for this article can be found online at http://dx.doi.org/10.7717/peerj-cs.1160#supplemental-information.

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
