# Peer review of "A new approach for SPN removal: nearest value based mean filter"

_PeerJ Computer Science, doi:10.7717/peerj-cs.1160_

## Round 0.1 · original submission · Major Revisions

Authors must take into account the suggestions made by the reviewers, especially regarding the inclusion of statistical validation for a comparison vs the state of the art, which is mandatory in this case. The paper will not be accepted otherwise.

Reviewer 2 has requested that you cite specific references. However, these citations do not appear to be relevant to your manuscript. You may add them if you believe they are relevant, but I do not expect you to include these citations, and if you do not include them, this will not influence my decision

Reviewer 1 ·

Basic reporting

This manuscript proposes a SPN filter method based on the nearest pixel values
Manuscript is written in well-structured manner. All the claims are justified with the help of logical and mathematical analysis. All the results are compared with other existed models. The algorithm and flowchart of the application are added. The image datasets used in the application are appropriate and sufficient.
It is appropriate this manuscript be accepted for publication.

Experimental design

The algorithm used in the study is supported by mathematical equations.

Experimental study was carried out using image datasets that are widely used in the literature. In this respect, the experimental design is appropriate.

Validity of the findings

Findings were compared with state-of-the-art SPN removal methods. High success rates were obtained with the proposed method in the study.

Reviewer 2 ·

Basic reporting

- English writing and presentation style should be improved. There contained grammatical errors, typos, or jargon.
- Quality of figures/tables should be improved significantly. For example, from Table 5 (it is figure as I think), some contents are missing, resolution must be improved.
- Lacking of the results. Currently, the authors focuses a lot on the discussions, but they should add some thing in the "Results" part.

Experimental design

- Statistical tests should be conducted when comparing the performance among methods/algorithms to see significant differences.
- Uncertainties of model should be reported.
- Some references related to image-based model i.e., PMID: 35648374, PMID: 34502160 should be added to attract a broader readership.
- The study is conducted on old datasets. Can they apply to some new data?

Validity of the findings

- The improvements are not much compared to previous methods.

Additional comments

No comment

---

## Round 0.2 · Major Revisions

Although two reviewers are pleased with the updated content of the manuscript, there are also several concerns raised by one of the reviewers in accordance with the novelty of the method and the comparison with respect to the state of the art.

Reviewer 1 ·

Basic reporting

This manuscript proposes a SPN filter method based on the nearest pixel values
Manuscript is written in well-structured manner. All the claims are justified with the help of logical and mathematical analysis. All the results are compared with other existed models. The algorithm and flowchart of the application are added. The image datasets used in the application are appropriate and sufficient.
It is appropriate this manuscript be accepted for publication.

Experimental design

The algorithm used in the study is supported by mathematical equations.
Experimental study was carried out using image datasets that are widely used in the literature. In this respect, the experimental design is appropriate.

Validity of the findings

Findings were compared with state-of-the-art SPN removal methods. High success rates were obtained with the proposed method in the study.
In addition, the article was supported by statistical tests at the revision stage.

Reviewer 2 ·

Basic reporting

No comment.

Experimental design

No comment.

Validity of the findings

No comment.

Additional comments

No comment.

Reviewer 3 ·

Basic reporting

In this paper, the author proposed a nearest value based mean filter method for salt and pepper noise removal. The two steps are designed: 1-Changing the noisy pixel value with the closest pixel value or assigning their average to the noisy pixel in case there is more than one pixel with the same distance. 2-It is the updating of the calculated noisy pixel values with the average ûlter by correlating them with the noise ratio. The paper is written well and easy to follow.

Experimental design

The comparisons are out-dated, more new methods should be compared, also color images should be tested. In Fig 3. and 4, compared with (I), i cannot see the improvements.

Validity of the findings

The method is outdated including the referenced papers. The contribution is also limited.

---

## Round 0.3 · accepted · Accept

Authors have successfully addressed the comments from the reviewers, and therefore the paper is now ready for publication.

Reviewer 2 ·

Basic reporting

My previous comments have been addressed.

Experimental design

My previous comments have been addressed.

Validity of the findings

My previous comments have been addressed.

Additional comments

My previous comments have been addressed.

Reviewer 3 ·

Basic reporting

all my comments have been addressed, i recommend accepting the paper

Experimental design

all my comments have been addressed, i recommend accepting the paper

Validity of the findings

all my comments have been addressed, i recommend accepting the paper

Additional comments

no